

# Estimation of refractivity uncertainties and vertical error correlations in collocated radio occultations, radiosondes and model forecasts

Johannes K. Nielsen[1], Hans Gleisner[1], Stig Syndergaard[1], and Kent B. Lauritsen[1]

[1]Danish Meteorological Institute
[1]Lyngbyvej 100, DK 2100, Copenhagen, Denmark

**Correspondence:** Johannes K. Nielsen (jkn@dmi.dk)

**Abstract.**

Random uncertainties and vertical error correlations are estimated for three independent data sets. The three collocated data sets are: 1) Refractivity profiles of radio occultation measurements retrieved from the Metop-A and B and COSMIC-1 missions, 2) refractivity derived from GRUAN processed RS92 sondes and 3) refractivity profiles derived from ERA5 forecast

fields. The analysis is performed using a generalization of the so-called Three-Cornered Hat method to include off-diagonal elements such that full error covariance matrices can be calculated. The impacts from various sources of representativeness error on the uncertainty estimates are analyzed. The estimated refractivity uncertainties of radio occultations, radiosondes and model data are stated with reference to the vertical representation of refractivity in these data sets. The existing theoretical estimates of radio occultation uncertainty are confirmed in the middle and upper troposphere and lower stratosphere, and only

little dependence on latitude is found in that region. In the lower troposphere refractivity uncertainty decreases with latitude. These findings have implications for both retrieval of tropospheric humidity from radio occultations and for assimilation of radio occultation data in NWP models and reanalyses.

## 1 Introduction

In variational estimation of geophysical parameters from satellite observations, the obtained accuracy relies on the validity

of the underlying uncertainty and error correlation assumptions of the observation and of the model background fields. The *Three-Cornered Hat* (3CH) method (Grubbs, 1948; Barnes, 1966; Levine, 1999) provides an empirically based uncertainty estimate of three independent data sets, all representing a series of measurements of the same physical property. A historical overview of the applications of 3CH, and related methods, is given by Sjoberg et al. (2020). The 3CH method was introduced independently by multiple authors earliest by Grubbs (1948), and (often referenced) Gray and Allan (1974). The method has

in several cases been used for meteorological applications, sometimes under other names, see e.g., O'Carroll et al. (2008).

In Numerical Weather Prediction (NWP), the method developed by Desroziers et al. (2005) is being widely adopted to empirically based adjustment of observation error covariance matrices, e.g., Bormann et al. (2016). However, the 3CH method has not been adopted as a tool in operational assimilation of satellite data into NWP models. This is likely because of the





requirements — that the errors of the three data sets must be uncorrelated, and that the data sets must truly represent the same
property with the same footprint in time and space — that are seldom met.

To distinguish error correlations between data sets from vertical error correlations within each data set, we will refer to
the former as *error cross correlations*. Such error cross correlations can for instance be due to similarities in measurement
methods and processing or they can for example arise as a result of similarity in resolution among the data sets. Error cross
correlations can cause the 3CH method to misrepresent uncertainties (Rieckh and Anthes, 2018). If error cross correlations
and representativeness issues are properly considered and accounted for, the 3CH method can serve as an alternative to — or a
validation reference for — uncertainty estimates based on instrument characteristics and measurement geometry.

Recently Rieckh et al. (2021) applied the 3CH method to refractivity, temperature and humidity profiles from radio occulta-
tions (RO), combined with radiosondes and model analysis. The results of that study gives relatively large uncertainty estimates
(see discussion Sect. 5), and the study leaves the problem of error cross correlations unresolved.
In this paper the 3CH method is generalized to include off-diagonal elements of the error covariance matrices. We apply
the generalized 3CH (G3CH) to three data sets where the random error components can be assumed to be truly independent.
The refractivity error covariance matrices of RO measurements are estimated and compared to current vertical correlation
assumptions, used in 1D-Var retrieval of specific humidity and temperature from RO refractivity. The main objective of this
study is to assess refractivity random uncertainty and vertical error correlations, expressed as the refractivity error covariance
matrix, to be used in 1D-Var retrieval of temperature and specific humidity (Healy and Eyre, 2000; Kursinski et al., 2000; ROM
SAF, 2021). The analysis will also provide estimates of the ERA5 refractivity error covariance matrix and of the GRUAN
processed RS92 refractivity error covariance matrix.

The rest of the paper is organized as follows: The next two sections, Sect. 1.1 and Sect. 1.2 contain definitions of the
terminology used throughout the paper. Next the three data sets are introduced in Sect. 2, and the G3CH method is presented
in Sect. 3, which includes a derivation of the G3CH equations. Results are presented in Sect. 4 along with interpretation of
the different collocation and filtering experiments. In the Discussion, Sect. 5, the results are related to previous studies and
applications. The results are finally collected in the Conclusions, Sect. 6.

## 1.1 Definitions

The terms *random uncertainty* and *systematic uncertainty* are used as defined in the Guide to the Expression of Uncertainty
in Measurement (GUM) (International Bureau of Weights and Measures and International Organization for Standardization,
1993). However, since the GUM does not provide a terminology for non-scalar properties we adopt the concept of *error covari-
ance (matrix)* and *error correlation (matrix)* from NWP terminology (Bormann et al., 2016) to describe vertically correlated
random uncertainties, and we use the terms *error variance* and *error standard deviation* to refer to the diagonal of an error
covariance matrix and its square root.
The term *vertical footprint* of a data set is used here in the meaning width of an ideal physical refractivity feature, shaped as
a delta function, mapped to the resolved representation of refractivity, for the given data set. The word *resolution* may be used
to describe this property, but we shall avoid this term because in the NWP community it is used in the meaning of sampling





density — the number of data points per spatial interval (for example in Hersbach et al. (2020)). The vertical footprint will typically be larger than the distance between data points.

## 1.2 Error components

For a given refractivity data profile, $\mathbf{x}$, we consider the observation error $\boldsymbol{\varepsilon}$ as the deviation from the unknown truth $\mathbf{t}$, $\boldsymbol{\varepsilon} = \mathbf{x} - \mathbf{t}$. In the context of this paper $\mathbf{t}$ is the actual refractivity at a vertical line above the RO reference coordinates at the RO reference time, defined with respect to given finite footprints in space and time, which may differ from the footprints of all three data sets.

The quantity $\boldsymbol{\varepsilon}$ is a sum of the measurement error $\boldsymbol{\varepsilon}^I$ and a representativeness error $\boldsymbol{\varepsilon}^R$. Both terms may contain random and systematic error components, but we assume that the systematic components have been removed prior to this analysis. The measurement error $\boldsymbol{\varepsilon}^I$ acts as a superimposed noise, possibly correlated in space and time. $\boldsymbol{\varepsilon}^I$ may for instance include instrument errors, radio noise from external sources, and also some errors arising during data processing steps. The $\boldsymbol{\varepsilon}^R$ component represents the distortion of the underlying truth in a data set, as it is being mapped to the observation grid. $\boldsymbol{\varepsilon}^R$ contains errors associated with for instance sampling, interpolation and mismatch between the observation grid and measurement resolution in time and space. Especially for the RO and RS92 data $\boldsymbol{\varepsilon}^R$ contains a geometric error component, $\boldsymbol{\varepsilon}^G$, representing the departure of the RO and RS92 trajectories in time and space from the vertical profile at the RO reference time. The ERA5 profile is strictly vertical, interpolated to the RO reference time and position, while the RO profile is a weighted average of the three-dimensional atmosphere in the plane of occultation Syndergaard et al. (2005)], and the radiosonde follows the balloon trajectory. The time scale of an RO profile is in the order of one minute and the timescale of an radiosonde profile is in the order of one hour. The skew trajectories of the RO tangent points and RS92 balloons are assumed not to be correlated with each other. Hence the $\boldsymbol{\varepsilon}^G$ term contains no cross correlations, and consequently it will be correctly attributed to the RO and RS92 data by the G3CH procedure.

Given the definition of $\mathbf{t}$ to be the actual profile at the RO reference location and time, there are, in addition to $\boldsymbol{\varepsilon}^R$ and $\boldsymbol{\varepsilon}^I$, errors induced by the methods applied in this paper. These are a collocation error, $\boldsymbol{\varepsilon}^C$, due to the distance in time and space between the radiosonde and the reference coordinates, and a cross-correlation error, $\boldsymbol{\varepsilon}^X$, representing error cross correlations induced by the finite footprints of the three data sets. The raw G3CH uncertainty estimate for one of the data sets will not represent the intrinsic error, but it will represent a combination of the intrinsic error $\boldsymbol{\varepsilon}$ and error components added by the G3CH:

$$\boldsymbol{\varepsilon}_{\mathrm{G3CH}} = \boldsymbol{\varepsilon}^I + \boldsymbol{\varepsilon}^R + \boldsymbol{\varepsilon}^C + \boldsymbol{\varepsilon}^X. \tag{1}$$

We are able to remove $\boldsymbol{\varepsilon}^C$ and the $\boldsymbol{\varepsilon}^X$ components of the three data sets, by adding additional analysis steps to the G3CH (see sections Sect. 4.2 and Sect. 4.3). So the observation error covariance matrices that we estimate includes measurement error $\boldsymbol{\varepsilon}^I$ and representativeness error $\boldsymbol{\varepsilon}^R$.

$$\boldsymbol{\varepsilon} = \boldsymbol{\varepsilon}^I + \boldsymbol{\varepsilon}^R. \tag{2}$$





The final estimate of $\varepsilon$ will be stated with reference to a common vertical footprint of the three data sets, which is determined by the data set with the largest footprint, ERA5. These general definitions of measurement error and representativeness error are thought to be applicable for all three data sets.

## 2  Data

Three data sets are combined in the analysis. The radio occultation dataset (RO) includes refractivity profiles from the Metop
and COSMIC-1 missions (Gleisner et al., 2020). These are downloadable as part of the ROM SAF CDR v1 and ICDR v1 data sets. The CDR v1 (Gleisner et al., 2021a) consists of RO data from several satellite missions data that have been reprocessed by the ROM SAF, using lower-level input data from both EUMETSAT and UCAR as input. The ICDR v1 (Gleisner et al., 2021b) consists of RO data from the Metop mission, that has been reprocessed by the ROM SAF, using input data from EUMETSAT. Secondly the radiosondes (RS92) are taken from the RS92-GDP.2 data set, provided by the GCOS Reference
Upper-Air Network, GRUAN, (Dirksen et al., 2014; Sommer et al., 2012). From these two data sets a collocated subset has been selected, from the criterion that the GRUAN central time and position must be within three hours and 300 km from the radio occultation reference point. In effect this ensures that the location criteria are met in the upper troposphere while measurements can be sampled further apart at both higher and lower altitude. The RO data has been subject to the ROM SAF quality control described in Steiner et al. (2020), and the GRUAN data has been pruned for a few extreme outliers. The third data set (ERA5)
is model forecast from the ERA5 data set (Hersbach et al., 2020) on model levels, retrieved from the ECMWF MARS archive. The forecast verification time has in each case been chosen such that the radio occultation has not been within the assimilation window used for initialization of the given forecast. The ERA5 forecast is prepared at model levels and interpolated in time (three hour grid) and horizontal space (1x1 deg grid) to the RO reference points. These interpolated ERA5 profiles are also provided as part of the ROM SAF CDR v1 and ICDR v1 data sets. The data spans a time interval from 2006 to 2020. A total
of 15597 collocations were found for this analysis. The ERA5 and RS92 data have been interpolated to the RO vertical grid of 247 levels (Lewis, 2009) with cubic splines.

## 3  Method

### 3.1  The Generalized Three-Cornered Hat method

The 3CH method has historically been applied to triplets of data without considering vertical error correlations, meaning that
the data sets have effectively been treated as scalar properties (Sjoberg et al., 2020). A straight forward generalization of the method allows us to also infer internal error correlations for each data set. In the Generalized 3CH (G3CH) it is assumed that we have three independent variables $\mathbf{x}, \mathbf{y}$ and $\mathbf{z}$, that are composed of four stochastic vectors; the truth $\mathbf{t}$, and three independent



error terms $\boldsymbol{\varepsilon}_x$, $\boldsymbol{\varepsilon}_y$ and $\boldsymbol{\varepsilon}_z$, such that

$$
\begin{aligned}
\mathbf{x} &= \mathbf{t} + \boldsymbol{\varepsilon}_x \\
\mathbf{y} &= \mathbf{t} + \boldsymbol{\varepsilon}_y \\
\mathbf{z} &= \mathbf{t} + \boldsymbol{\varepsilon}_z.
\end{aligned}
\tag{3}
$$

In the present paper $\mathbf{x}, \mathbf{y}$ and $\mathbf{z}$ may represent atmospheric refractivity profiles obtained from different sources. $\boldsymbol{\varepsilon}_x, \boldsymbol{\varepsilon}_y$ and $\boldsymbol{\varepsilon}_z$ represent the random observation error vectors. In the following the bracket notation, $\langle \cdot \rangle$, is used to denote expectation values.

The error vectors $\boldsymbol{\varepsilon}_x, \boldsymbol{\varepsilon}_y$ and $\boldsymbol{\varepsilon}_z$ may have internal correlations, expressed as error covariance matrices $\mathbf{X} = \langle \boldsymbol{\varepsilon}_x \boldsymbol{\varepsilon}_x{}^T \rangle$, $\mathbf{Y} = \langle \boldsymbol{\varepsilon}_y \boldsymbol{\varepsilon}_y{}^T \rangle$ and $\mathbf{Z} = \langle \boldsymbol{\varepsilon}_z \boldsymbol{\varepsilon}_z{}^T \rangle$, but we assume no cross correlation components, that is; $\langle \boldsymbol{\varepsilon}_x \boldsymbol{\varepsilon}_y^T \rangle = \langle \boldsymbol{\varepsilon}_x \boldsymbol{\varepsilon}_z^T \rangle = \langle \boldsymbol{\varepsilon}_z \boldsymbol{\varepsilon}_y^T \rangle = 0$. We may allow that the error is correlated with the physical property $\mathbf{t}$; e.g., $\langle \mathbf{t} \boldsymbol{\varepsilon}_x^T \rangle \neq 0$. In the analysis in the present paper we only estimate the random uncertainties, so it can without loss of generality be assumed that all three data sets are bias free. This can in practice be ensured by subtracting the mean of each data set prior to the analysis. In the absence of bias the covariance matrices of each subtraction pair can be written as

$$
\begin{aligned}
\langle (\mathbf{x} - \mathbf{y})(\mathbf{x} - \mathbf{y})^T \rangle &= \langle \mathbf{x}\mathbf{x}^T + \mathbf{y}\mathbf{y}^T - \mathbf{x}\mathbf{y}^T - \mathbf{y}\mathbf{x}^T \rangle \\
\langle (\mathbf{x} - \mathbf{z})(\mathbf{x} - \mathbf{z})^T \rangle &= \langle \mathbf{x}\mathbf{x}^T + \mathbf{z}\mathbf{z}^T - \mathbf{x}\mathbf{z}^T - \mathbf{z}\mathbf{x}^T \rangle \\
\langle (\mathbf{y} - \mathbf{z})(\mathbf{y} - \mathbf{z})^T \rangle &= \langle \mathbf{y}\mathbf{y}^T + \mathbf{z}\mathbf{z}^T - \mathbf{y}\mathbf{z}^T - \mathbf{z}\mathbf{y}^T \rangle.
\end{aligned}
\tag{4}
$$

Expanding the right hand side of for instance the first line of Eq. (4) we obtain:

$$
\langle (\mathbf{x} - \mathbf{y})(\mathbf{x} - \mathbf{y})^T \rangle = \langle \boldsymbol{\varepsilon}_x \boldsymbol{\varepsilon}_x{}^T - \boldsymbol{\varepsilon}_x \boldsymbol{\varepsilon}_y^T - \boldsymbol{\varepsilon}_y \boldsymbol{\varepsilon}_x^T + \boldsymbol{\varepsilon}_y \boldsymbol{\varepsilon}_y{}^T \rangle.
\tag{5}
$$

If we keep in mind that error cross correlations between data sets are set to zero, the three subtraction pair covariances reduces to

$$
\begin{aligned}
\langle (\mathbf{x} - \mathbf{y})(\mathbf{x} - \mathbf{y})^T \rangle &= \langle \boldsymbol{\varepsilon}_x \boldsymbol{\varepsilon}_x{}^T + \boldsymbol{\varepsilon}_y \boldsymbol{\varepsilon}_y{}^T \rangle \\
\langle (\mathbf{x} - \mathbf{z})(\mathbf{x} - \mathbf{z})^T \rangle &= \langle \boldsymbol{\varepsilon}_x \boldsymbol{\varepsilon}_x{}^T + \boldsymbol{\varepsilon}_z \boldsymbol{\varepsilon}_z{}^T \rangle \\
\langle (\mathbf{y} - \mathbf{z})(\mathbf{y} - \mathbf{z})^T \rangle &= \langle \boldsymbol{\varepsilon}_y \boldsymbol{\varepsilon}_y{}^T + \boldsymbol{\varepsilon}_z \boldsymbol{\varepsilon}_z{}^T \rangle.
\end{aligned}
\tag{6}
$$





Finally, by solving these three equations for the error covariance matrices $\mathbf{X} = \langle \boldsymbol{\varepsilon}_x \boldsymbol{\varepsilon}_x{}^T \rangle$, $\mathbf{Y} = \langle \boldsymbol{\varepsilon}_y \boldsymbol{\varepsilon}_y{}^T \rangle$ and $\mathbf{Z} = \langle \boldsymbol{\varepsilon}_z \boldsymbol{\varepsilon}_z{}^T \rangle$ for the variables $\mathbf{x}, \mathbf{y}$ and $\mathbf{z}$, we get

$$
\begin{aligned}
\mathbf{X} = \langle \boldsymbol{\varepsilon}_x \boldsymbol{\varepsilon}_x{}^T \rangle &= \frac{1}{2}\langle (\mathbf{x} - \mathbf{y})(\mathbf{x} - \mathbf{y})^T + (\mathbf{x} - \mathbf{z})(\mathbf{x} - \mathbf{z})^T \\
&\quad - (\mathbf{z} - \mathbf{y})(\mathbf{z} - \mathbf{y})^T \rangle \\
\mathbf{Y} = \langle \boldsymbol{\varepsilon}_y \boldsymbol{\varepsilon}_y{}^T \rangle &= \frac{1}{2}\langle (\mathbf{y} - \mathbf{x})(\mathbf{y} - \mathbf{x})^T + (\mathbf{y} - \mathbf{z})(\mathbf{y} - \mathbf{z})^T \\
&\quad - (\mathbf{x} - \mathbf{z})(\mathbf{x} - \mathbf{z})^T \rangle \\
\mathbf{Z} = \langle \boldsymbol{\varepsilon}_z \boldsymbol{\varepsilon}_z{}^T \rangle &= \frac{1}{2}\langle (\mathbf{z} - \mathbf{x})(\mathbf{z} - \mathbf{x})^T + (\mathbf{z} - \mathbf{y})(\mathbf{z} - \mathbf{y})^T \\
&\quad - (\mathbf{x} - \mathbf{y})(\mathbf{x} - \mathbf{y})^T \rangle.
\end{aligned}
\tag{7}
$$

The above G3CH model, is applied to the three data sets described in Sect. 2. In this analysis the mean is subtracted from each data set prior to applying the G3CH. The biases are not the focus here, but for reference the global means of RS92 and RO refractivity differences to ERA5 for all collocations used in the analysis are plotted in Fig. 1.

### 3.2 Handling collocation uncertainty

In order to compensate for the impact of collocation uncertainty $\varepsilon^C$ on the obtained refractivity error covariance matrices, the G3CH analysis is applied to a series of data subsets with increasing collocation distances between 50 km and 300 km. The collocation uncertainty is removed from the uncertainty estimates by extrapolating the covariance matrices to zero collocation distances. This procedure, which is also performed by Hollingsworth and Lönnberg (1986) in another context, also allows one to track how the G3CH method partition the collocation uncertainty among the three data sets. See subsection 4.2.

### 3.3 Error correlations between data sets

The 3CH algorithm cannot distinguish between true physical variability and mutual positive error correlations (Sjoberg et al., 2020). In cases where errors of two data sets ($\mathbf{x}$ and $\mathbf{y}$) are positively correlated the discrepancy between the third dataset, $\mathbf{z}$ and $(\mathbf{x}, \mathbf{y})$ will be attributed as an uncertainty of $\mathbf{z}$, because the term $\langle (\mathbf{x} - \mathbf{y})(\mathbf{x} - \mathbf{y})^T \rangle$ would be reduced in such cases.

In this study the measurement error cross correlations between the chosen data sets are assumed to be negligible, since the three data sets at hand are obtained by completely independent techniques. In particular the ERA5 model forecast data is chosen such that no information from either a given RO or RS92 profile can have been passed to the forecast being used in a given collocation triplet. However, if two data sets have similar vertical footprints, differing from the vertical footprint of $\mathbf{t}$, these two data sets will have cross-correlated errors, and possibly biases. All biases are removed prior to application of G3CH, but the error cross correlations introduced by finite vertical footprints will influence the result of G3CH.

### 3.4 Handling differences in vertical footprints

The three data sets differ in their vertical footprints. The RS92 radiosonde has a vertical footprint of around 50 m (Dirksen et al., 2014). This footprint is increased through the interpolation to the common grid, and through the procedure for correcting





for collocation error. The radio occultation refractivity has been shown to have a vertical footprint of about 200 m under optimal conditions in the lower troposphere (Xie et al., 2012). In the RO data used here the processing has removed some small scale information, so the RO vertical footprint is expected to be larger than 200 m. In Fig. 2 two examples of refractivities of triple-collocations are shown. The plots illustrate the ability of resolving vertical structures in the middle troposphere and lower stratosphere of the three data sets. Even though the highly resolved ERA5 has 137 vertical levels, shown on the right vertical axis, it provides a somewhat smoother representation of the vertical structures, compared to the radio occultation. The RO profiles and RS92 profiles, show more vertical structure than ERA5.

Uncertainty estimates for any variable must refer to a specified footprint to be meaningful. Thus, the G3CH analysis has to be accompanied with an assessment of the footprints of the data sets. In the three cornered hat analysis, the data set with the largest footprint determines the common footprint to be used for all three data sets. Said in another way: If one of the data sets do not contain information below a certain length scale, there is not enough information in the data triplet to apply the G3CH method to estimate uncertainties related to variability below that length scale.

Because ERA5 is missing some fine scale physical features, seen in the better resolved RO and RS92 data set, we are forced to state the uncertainty on the common scale determined by ERA5. This means that the RO and RS92 data must be smoothed to match the ERA5 footprint prior to the G3CH analysis. If this smoothing is omitted the G3CH may give a biased estimate of uncertainties. By smoothing the data sets to a common scale we remove both physical features and errors on scales shorter than the common footprint. Therefore the estimated uncertainties of RO and RS92, which are correct on the found common scale, may be viewed as lower uncertainty boundaries for these variables, on their native scales. The vertical footprints of the three data sets are examined in Sect. 4.3.

## 4 Results

In this section the G3CH results are presented, first as raw unfiltered uncertainty estimates, then with corrections for collocation mismatch ($\varepsilon^C$ terms) and corrections for cross correlations due to finite vertical footprints ($\varepsilon^X$ terms), to assess the uncertainty limits for each data set.

### 4.1 Raw uncertainty estimates

Fig. 3 shows the raw estimates of the mid latitude refractivity uncertainty expressed as error standard deviation of the three data sets, obtained by applying the G3CH directly to the raw data sets. Generally the G3CH attributes a big part of the collocation error ($\varepsilon_x^C$) to the RS92 uncertainty. The reason is that the collocation is performed by interpolating ERA5 to the RO reference point, such that ERA5 and RO are closely collocated, while RS92 is being chosen such that it is within 300 km from the RO reference point, so naturally RS92 will stand out from the two other data sets in many cases.



## 4.2 Collocation uncertainty

The most striking feature in Fig. 3 is the bulge of RS92 around the tropopause. The main part of this bulge is removed along with the collocation uncertainty by the procedure described in Sect. 3.2. We are calculating the G3CH estimates of covariance

matrices for a sequence of collocation criteria (between 50 km and 300 km) and use these to extrapolate all covariance matrices to 0 km collocation distance, with a linear fit to the squared collocation distance. The impact on RS92 of changing collocation distance is shown as an example in Fig. 4, and in Fig. 5 a few examples of extrapolations are shown. The result of this procedure is summarized for all three data sets in Fig. 6. The RS92 uncertainty estimate is reduced considerably, while the uncertainty estimates for the two other data sets are slightly changed. In the subsequent analysis the 0 km estimates of covariances are used

for evaluation of covariance matrices and difference terms in the G3CH equations.

## 4.3 Vertical filtering

In Fig. 7 the impact of smoothing on error standard deviations estimated with G3CH (Eq. 7) is shown for middle latitudes. The smoothing is applied as a sequence of Gaussian filters of increasing widths. For each data set filtering has been performed, not on the data set itself, but on the two other complementing data sets (see figure legends). The G3CH analysis has been

performed at the sequence of such prepared triplets of data sets with increasing filter width. The impact of applying sequences of Gaussian filters is best viewed near the tropopause. We note that all variances eventually starts to grow at some filter width, but the ERA5 error standard deviation drops in most cases at small filter widths, and does not start to increase until the width of the filter, applied on the RO and RS92 data, exceeds a certain threshold. We interpret this threshold as the ERA5 footprint. ERA5 footprint was estimated for each altitude, as the minimum of a second order polynomial, fitted to $\sigma_{\mathrm{ERA5}}$ as function

of filter width. These footprints are plotted in Fig. 8, for middle and high latitudes. At low latitudes the result is unstable, so that plot has been omitted. We use these result to identify a common ERA5 footprint to be applied globally as the mean of the middle and high latitude footprints, shown as a dashed line in Fig. 8.

A similar analysis cannot be performed for the RO or RS92 data sets, because these appear to have small footprints which happens to lie close to each other. There is not a finite filter length which minimizes the refractivity error standard deviation

for RO and RS92 (the filters being applied to the complementing data sets in each case). Therefore the RO and RS92 footprints cannot be inferred from these three data sets alone, but it can be concluded that their footprints are smaller than the ERA5 footprint since ERA5 estimated error standard deviation decreases if either the RO or RS92 are smoothed. This is illustrated in Fig. 9: The impact of smoothing RS92 on the ERA5 variance is shown in Fig. 9 (a). Generally $\sigma^2_{\mathrm{ERA5}}$ decreases as the RS92 data are brought closer to the ERA5 data by smoothing, consistent with RS92 having smaller footprint than ERA5. The RO

error variance, $\sigma^2_{\mathrm{RO}}$, on the other hand increases as a result of smoothing the RS92 data (see Fig. 9 (b)). This is consistent with the RS92 footprint being close to the RO footprint, and RS92 data moving closer to the ERA5 data as smoothing is applied to RS92. In Fig. 9 (c) $\sigma^2_{\mathrm{ERA5}}$ is seen to decrease as the RO refractivity is brought closer to ERA5 refractivity, as smoothing is applied on RO.



To estimate the final G3CH uncertainties with reference to the common footprint determined by ERA5, all three raw data sets have been smoothed with a Gaussian filter with the width of the ERA5 footprint prior to the G3CH analysis. In Fig. 10 the final G3CH inferred uncertainties are shown for each data set for low, middle and high latitudes. For all data set the unfiltered (raw) uncertainty is also plotted, for later discussion.

## 4.4 Error covariances

In Fig. 11 and Fig. 12 the G3CH based error covariance matrices for ERA5, RO and RS92 are shown for rising and setting occultations for middle and high latitudes. These matrices have been calculated without any vertical filtering applied. The tropics are not shown because of insufficient amount of data in that region. The fine scale off-diagonal structures must be attributed to statistical noise, but there are certainly larger scale vertical correlation structures especially in the RO and RS92 data.

Generally the vertical correlations are divided in two separable regimes: Close to the diagonal we see a short range correlation with standard deviation of approximately 0.5 km, and a long range correlation component of varying shape and amplitude. The short scale vertical correlations are very similar for all data sets. Rising occultations are found to have larger vertical error correlations (and slightly larger standard deviation where correlations are broader) than setting occultations in this data set, which is seen when comparing plot (b) with plot (e) in Fig. 11 and plot (b) with plot (e) in Fig. 12. This is believed to be due to the ionospheric correction in the RO processing for rising occultations, where the L2 GPS signal is often not available below 20 km, and extrapolation from above is necessary. In the CDR v1.0 data set, it is in particular the rising occultations for Metop after instrument firmware upgrades in 2013 that suffers from missing L2 data below 20 km (Gleisner et al., 2020), and consequently there are broader vertical error correlations in the retrieved refractivity profiles for Metop after 2013 (not shown).

It is worth noticing that the estimated vertical correlations of RS92 are larger for setting than for rising RO at high latitudes, especially between 6 and 22 km. So the G3CH fails to give an independent estimate of the RS92 correlations. The RS92 is expected to have long ranging vertical correlations due to corrections implemented in the GRUAN processing, but the G3CH fails to attribute these correctly when strong long range correlations are also present in the RO data. The estimated RS92 diagonals (standard deviations superimposed vertically on correlation matrices) seem reasonably consistent for rising and setting occultations.

The relative magnitude of the off-diagonal covariance components can also be viewed in Fig. 13: Here the vertical error correlation function of RO refractivity is exemplified for two heights, approximately 5 and 20 km, at low, middle and high latitudes. The correlation functions are slices of the RO refractivity error correlation matrix at these altitudes. For instance at high latitude there are pronounced long range correlations at these two heights. In the tropics the data are too sparse to get an estimate of the correlation function. In Fig. 13 the dashed lines shows the three km exponential correlation which is assumed in the current ROM SAF 1D-Var analysis (ROM SAF, 2021). Given that the finer correlation structures, around the 1 km scale, are influenced by sparseness of data, the current correlation function appears to be reasonably adequate at high latitudes at the selected altitudes. At middle latitudes there is a potential for decreasing the error correlation length in future applications.



## 5 Discussion

It is evident from the results in Sect. 4.3 that uncertainty estimates for a data set must be stated with reference to scale in space and time. In the derivation of G3CH representativeness is defined with reference to given scales in space and time of the

truth. The truth is assumed to have smaller footprint than any of the involved data sets. We choose for all data sets to report the estimated uncertainty boundaries with reference to the estimated footprint of the ERA5 data set. We have identified the footprint of ERA5 for for a range of altitudes, and used these to find RO uncertainties in Fig. 10. This operation is equivalent to define the truth $\mathbf{t}$ with reference to the ERA5 footprint if one will.

The unfiltered (raw) G3CH RO uncertainty estimates, also seen in Fig. 10 includes uncertainty associated with fluctuations on

shorter scale than the ERA5 footprint. The raw uncertainties are overestimated because they will include physical variability, falsely attributed as errors. We cannot quantify the native footprints of RO and RS92, but it can be assumed that the raw uncertainty estimates marks upper boundaries for their uncertainties evaluated with reference to the native footprint.

The estimated uncertainties of RO in the lower stratosphere are only slightly above 0.2%, the theoretical estimates found in Kursinski et al. (1997). Empirical uncertainty estimates have previously been analysed for instance by Rieckh and Anthes

(2018) and Scherllin-Pirscher et al. (2011). In the empirical uncertainty analysis study by Scherllin-Pirscher et al. (2011) the refractivity uncertainty is found to be 0.35% in the lower stratosphere. Scherllin-Pirscher et al. (2011) were using RO data in combination with ECMWF analysis, and were therefore confined to make assumptions about mutual correlation and partitioning of uncertainties among the two data sets. By combining three independent data sets in a 3CH analysis such assumptions may be avoided, and consequently one is able to decrease the uncertainty estimates.

The RO uncertainty in the Upper Troposphere — Lower Stratosphere (UTLS), where the uncertainty estimates are at a minimum, does not vary much with latitude, except for a small increase at the tropopause. The structure of the uncertainty profiles are quite similar for all latitudes. The increase of uncertainty in the troposphere is smaller at high latitude, but the crossover between high and low uncertainty happens at approximately the same altitude, 5 to 7 km, for all latitudes. For the tropics, the noisy uncertainty estimates, due to insufficient amount of data, does not allow to safely read a minimum above the

tropopause from Fig. 10, but there is an indication of uncertainty almost down to 0.2% even between 7 and 10 km, well below the tropical tropopause layer.

In Rieckh et al. (2021) the 3CH is applied to multiple triplets of RO data and different model refractivity data. Overall their uncertainty estimates are much higher, for instance 0.55% in the UTLS. Partly this may be due to high noise level of the analyzed RO data (COSMIC-1 and COSMIC-2). But the data-sets used by Rieckh et al. (2021) are less well suited for 3CH

analysis than the data-sets used here, because the errors of the used models must be expected to be correlated across data-sets, leading to a biased estimate of 3CH uncertainties. The models applied have larger footprint than the RS92 radiosonde data used here, and this will tend to yield an overestimate of RO uncertainties, which is indeed seen in Rieckh et al. (2021). Their results show an apparent uncertainty maxima near the tropopause, as must be anticipated according to our analysis of impact of vertical footprint in Sect. 4.3.





In the theoretical uncertainty analysis study by Kursinski et al. (1997) the refractivity uncertainty in the lower troposphere tangents 1% which is a little lower than the G3CH estimate at middle latitudes, but quite consistent with the G3CH estimate at high latitudes. The Kursinski et al. (1997)-estimate is dominated by the contribution from representativeness uncertainty arising from horizontal gradients along the ray path up to 30 km, i.e. the contribution from the spherical symmetry assumption to the uncertainty. Retrospectively seen Kursinski et al. (1997) may have underestimated the horizontal gradients in the troposphere

since the effect of horizontal variability was estimated from a model with 40 km horizontal resolution. In a later study by Steiner and Kirchengast (2005) even lower refractivity uncertainty estimates are obtained, with a coarser model (60 km resolution and 50 vertical levels). Steiner and Kirchengast (2005) find down to 0.1% refractivity uncertainty in the UTLS.

     The estimation of error correlation matrices with G3CH is a novelty introduced in the present study. The method is able to detect expected differences between rising and setting RO long range vertical error correlations, and long range correlations

are also seen in RS92 data. The vertical correlation estimates are limited in the same way as variance estimates; if for instance one of the data-sets does not have long range correlations (in the case ERA5) the method fails to give an unbiased estimate of long range vertical error correlations of the two other data sets. However, this inaccuracy seems relatively small compared to the difference between the found RO vertical error correlation estimates and vertical error correlation estimates currently used in RO 1D-Var retrievals, and therefore the correlation estimates will be useful in this context. Especially at middle latitudes

there is a potential for decreasing the vertical error correlation length in future applications.

     The presented analysis G3CH may have consequences for uncertainty parameterization in retrieval and assimilation of RO refractivity. In particular the estimates of RO refractivity uncertainty reveals a potential for deflating tropospheric refractivity uncertainty in the ROM SAF 1D-Var configuration. A reduction of assumed refractivity uncertainty is of particular interest in the troposphere, where it can improve the information content of water vapor retrievals. There is a need for establishing tropo-

spheric water vapor climate data records for climate research, as it is for instance expressed in the objectives of the GEWEX water vapor assessment (G-VAP) (Schröder et al., 2018). The results presented here promises a reduction of uncertainty of RO based tropospheric water vapor retrieval.

## 6   Conclusions

The collocation-corrected G3CH random uncertainty estimate provides full refractivity error covariance matrices, for three

independent data sets. The method was a applied to collocated refractivity profiles from ERA5 forecast, radio occultations and radiosondes.

     The RO refractivity uncertainty is found between 0.2% and 0.6% in the UTLS between 8 and 25 km at middle and high latitudes, and between 0.2% and 1.4% below 8 km. The Generalized 3CH method presented here also yields estimates of the vertical error covariance matrices for refractivity.

The achieved refractivity uncertainty estimates are lower than empirically determined uncertainties previously reported in literature. The results can be used to model uncertainty assumptions used in model data assimilation and in 1D-Var calculation of atmospheric temperature and specific humidity based on RO refractivity data.



*Code and data availability.* The analysis is performed in Jupyter Notebook. The code is sitting in an internal git repository at the Danish Meteorological Institute. It is available from the corresponding author upon request.

The RO refractivity profiles and interpolated ERA5 temperature and humidity profiles on model levels (including surface pressure) are available at the ROM SAF web-page (Gleisner et al. (2021a) and Gleisner et al. (2021b)). The GRUAN atmospheric profiles are available at the GRUAN web-page (Sommer et al. (2012)).

*Author contributions.* Johannes K. Nielsen has put together (and partly invented) the method, done the analysis and is the main author of the manuscript. Stig Syndergaard, Kent B. Lauritsen and Hans Gleisner has contributed with substantial reformulations of every single part of

the manuscript, through multiple iterations, which have indeed led to changes in the basic rationale of the paper.

*Competing interests.* There are, to our knowledge, no competing interests to this paper.

*Acknowledgements.* This work was carried out as part of EUMETSAT's Radio Occultation Meteorology Satellite Application Facility (ROM SAF) which is a decentralised operational RO processing center under EUMETSAT. J. K. Nielsen, H. Gleisner, S. Syndergaard, and K. B. Lauritsen are members of the ROM SAF.



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



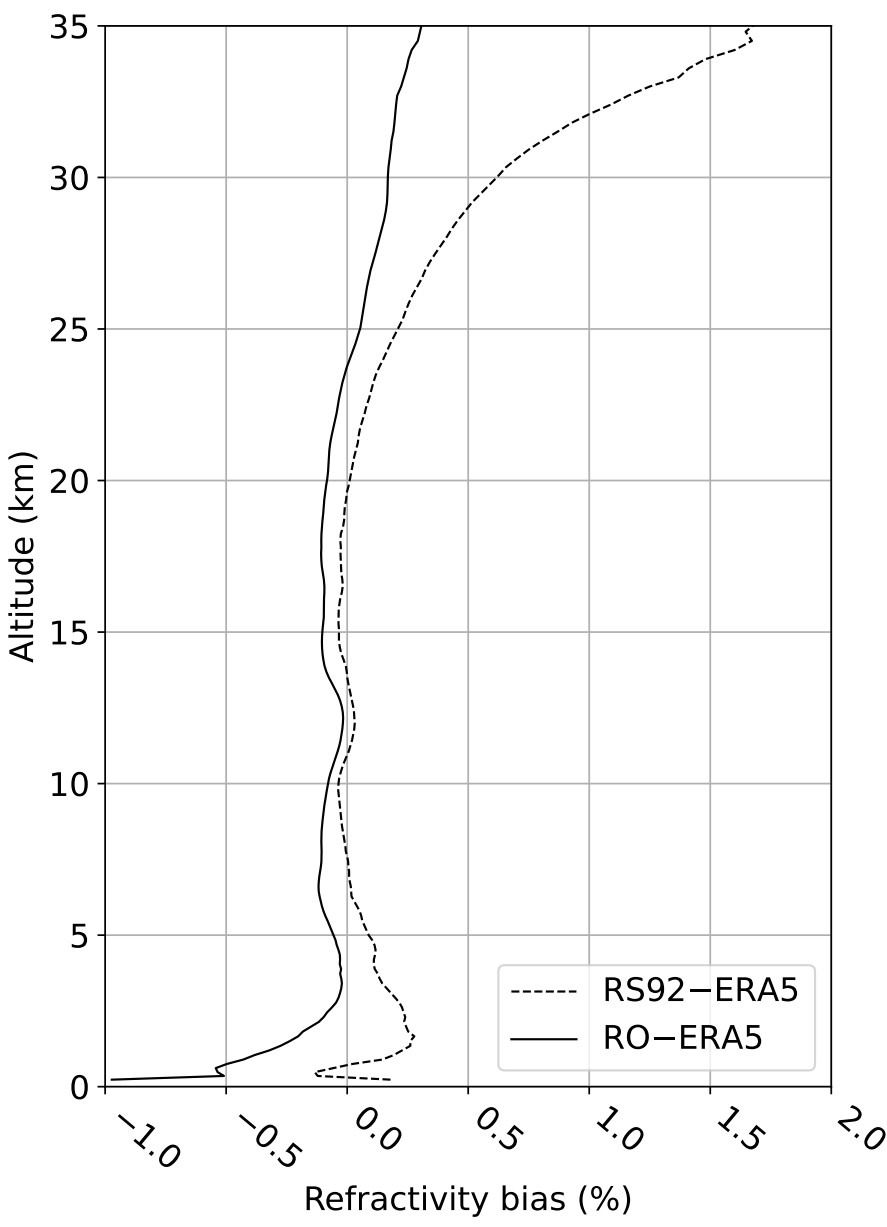

**Figure 1.** Global biases of RS92 and RO refractivity, with ERA5 used as reference, based on all collocation triplets (collocated ERA5, RO and RS profiles) used in this study, evaluated at the RO reference location. See also Sect. 2. Percentages are calculated relative to the mean of ERA5 forecasts.



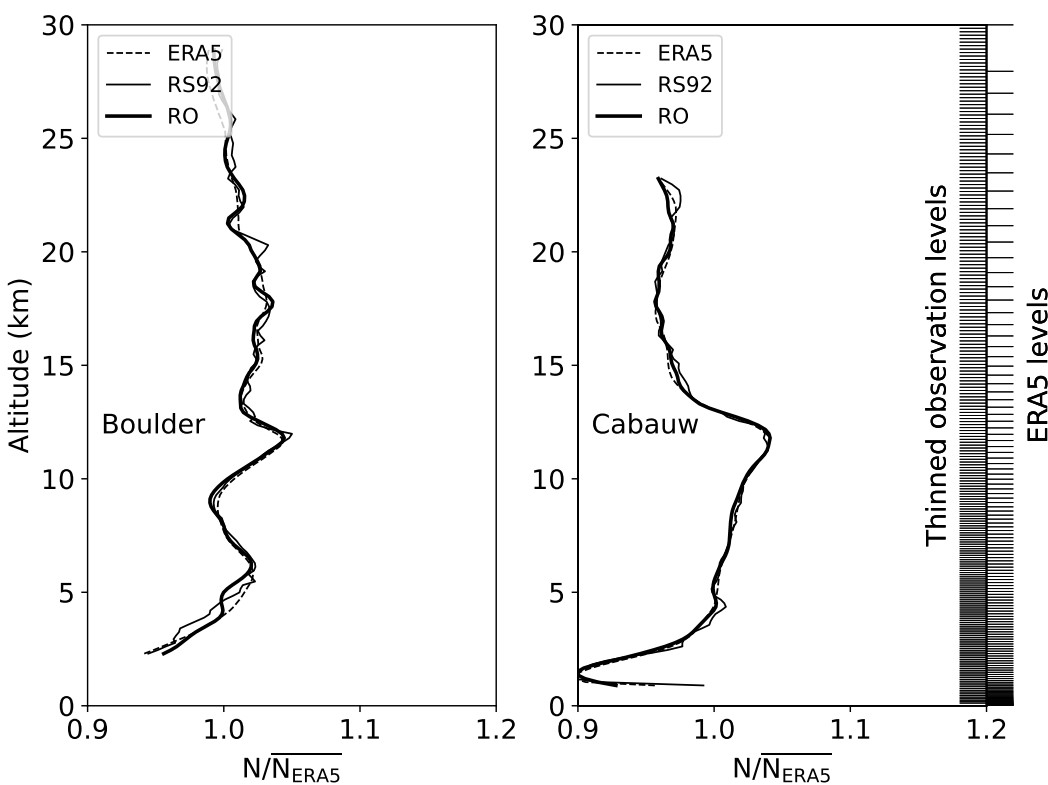

**Figure 2.** Refractivity of two selected triple collocations exemplifying differences in vertical footprint. Left: ERA5, RS92 and COSMIC-1 RO (40.1 deg N, collocation distance: 18.6 km (0.2 hour)) and right: ERA5, RS92 and Metop RO (52 deg N, collocation distance: 25.5 km (1.6 hour)). RS92 and ERA5 have been interpolated to the 247 RO height levels. The refractivity has been normalized by division with the mean of the ERA5 data set refractivity. The thinned refractivity levels and the 137 ERA5 model levels are printed on the right vertical axis.



**Figure 3.** Raw estimate of refractivity random uncertainty (standard deviation) of ERA5, RO and RS92 at middle latitudes.



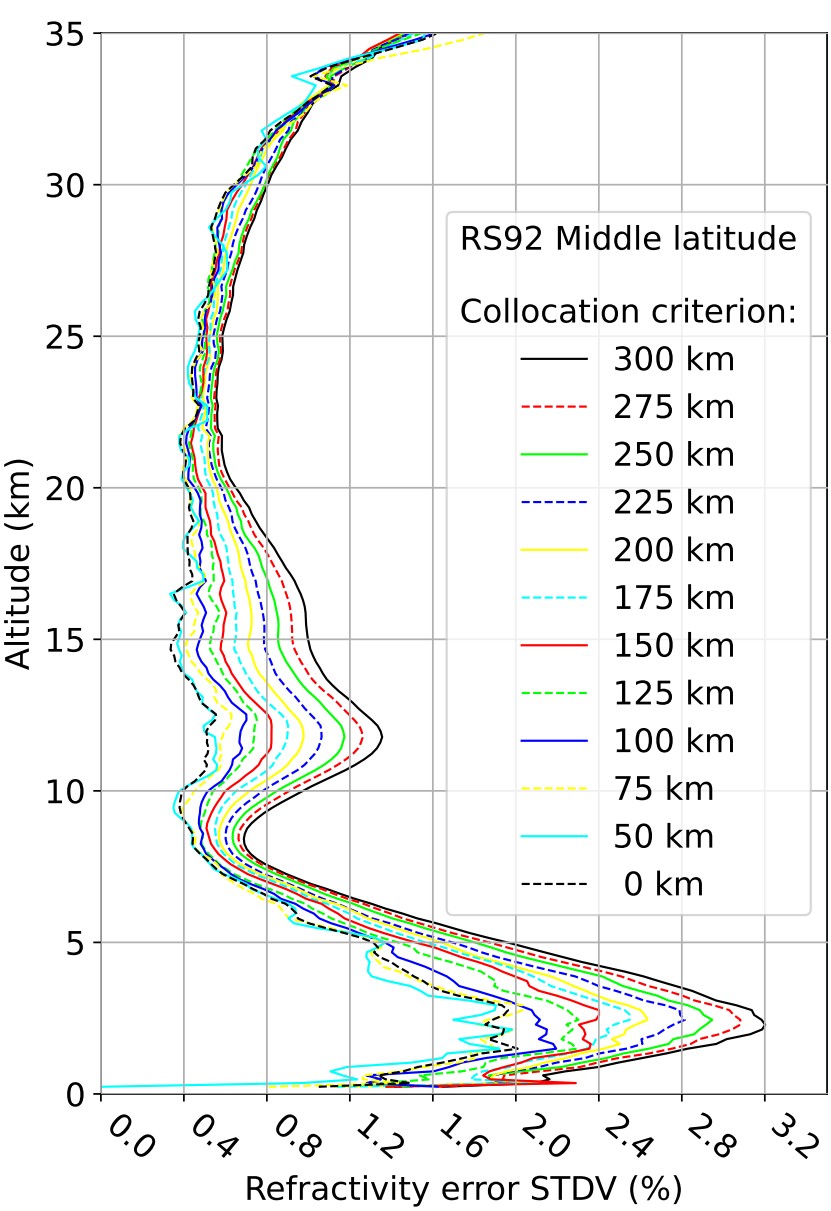

**Figure 4.** Estimate of refractivity uncertainty (standard deviation as percent of ERA5 mean refractivity) of RS92, for a series of collocation criteria. The black dashed line shows the STDV obtained by extrapolation of the variance to zero collocation distance.

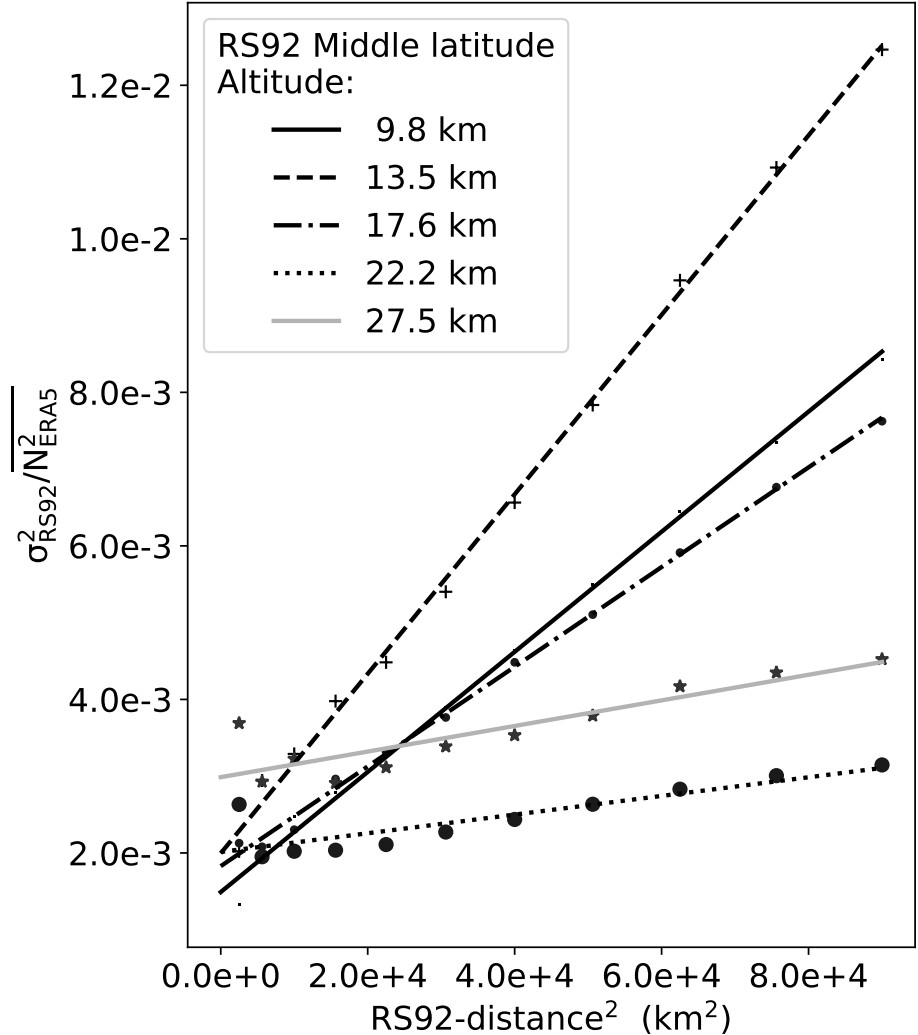

**Figure 5.** Examples of extrapolation of refractivity error variance of RS92, i.e. diagonal elements of covariance matrix, to zero collocation criterion at 5 different altitudes. The variances are divided with the mean ERA5 refractivity



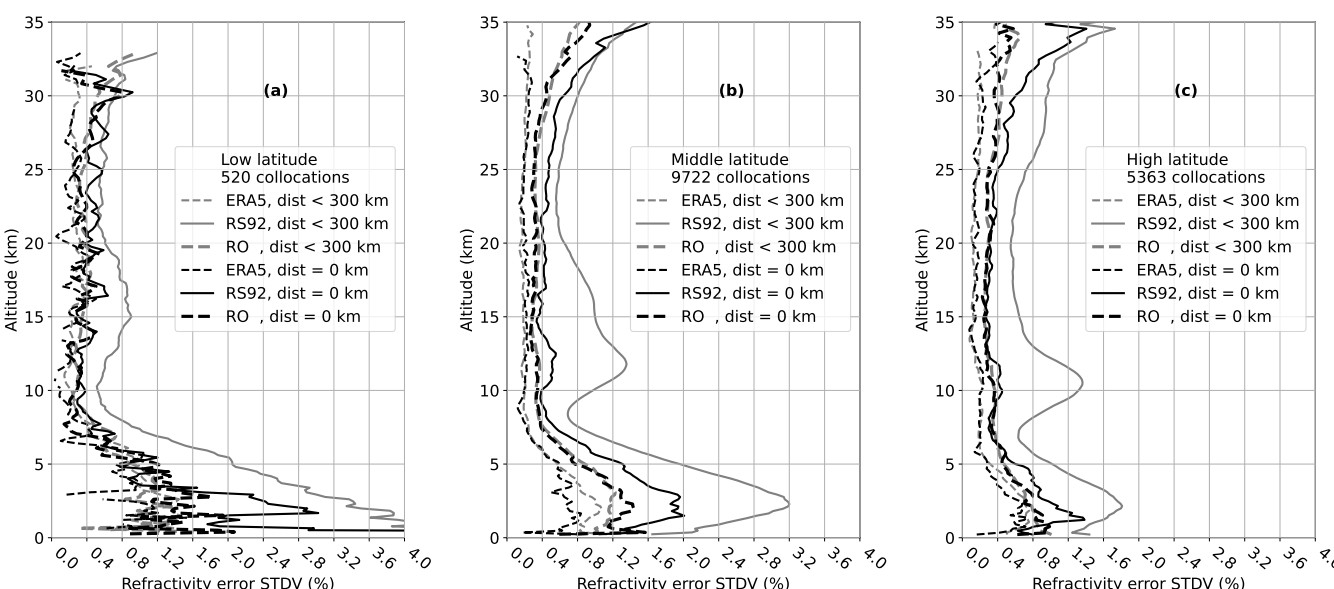

**Figure 6.** 3CH estimates of refractivity error standard deviations are shown for 300 km and 0 km collocation criterion, for each of the ERA5, RS92 and RO data sets, at low (a), middle (b) and high (c) latitudes. Smoothing has not been applied.



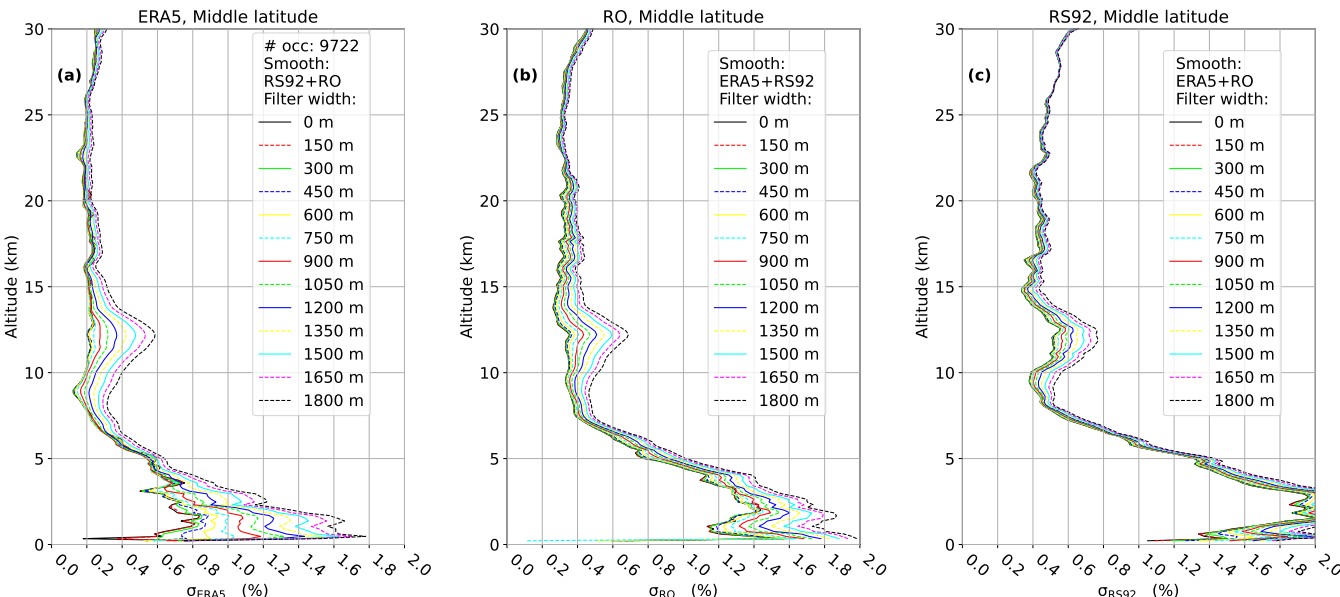

**Figure 7.** Effect of smoothing on error standard deviations of ERA5 (a), RO (b) and RS92 (c), estimated by 3CH. All plots are for middle latitudes. The standard deviations are divided with the mean ERA5 refractivity. The filtering width is defined as 2 times the standard deviation of the Gaussian filter function. The single curves are easiest identified at a given altitude by counting the curves from one side. For each data set the smoothing is only applied on the two other data sets.



**Figure 8.** Estimates of the ERA5 footprint for middle and high latitudes. The effect of filtering of RO and RS92 on the estimated ERA5 error standard deviation, $\sigma_{\mathrm{ERA5}}$, may be viewed in Fig. 7 (a). The ERA5 footprint is found for each altitude as the filter width which minimizes $\sigma_{\mathrm{ERA5}}$.



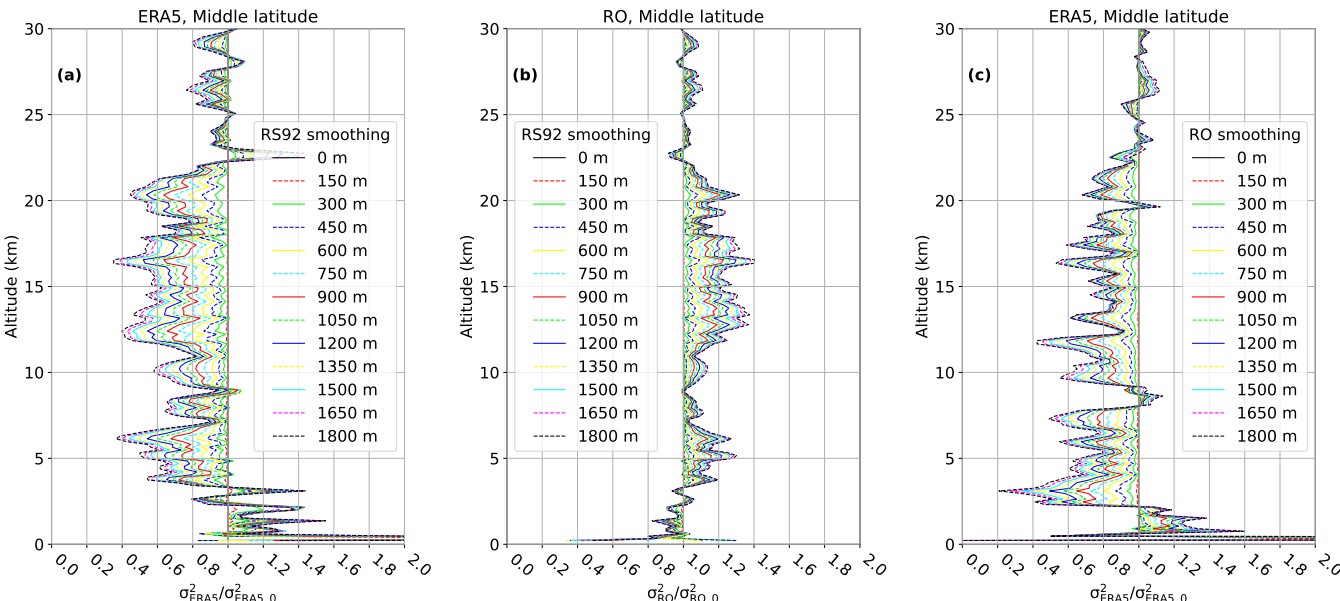

**Figure 9.** Estimates of middle latitude refractivity error variances based on smoothed data divided with refractivity error variance based on un-smoothed data; (a) ERA5 error variance with smoothed RS92, (b) RO error variance with smoothed RS92 and (c) ERA5 error variance with smoothed RO. The legends show the width of the different vertical Gaussian filters applied to the refractivity profiles mentioned in the legend title. 9722 collocated data triplets were used in the G3CH analysis for these error covariance estimates.

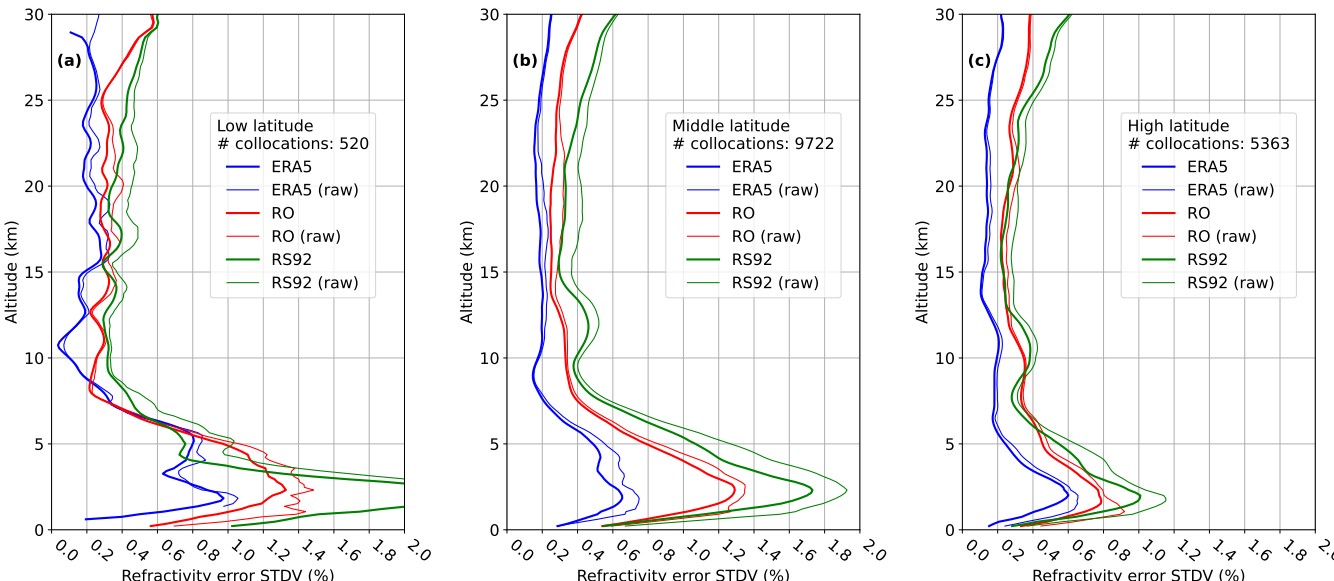

**Figure 10.** Best estimate of refractivity uncertainties, shown as standard deviations in percents of the ERA5 refractivity, for ERA5, RO and RS92, at low (a), middle (b) and high (c) latitudes. For all data sets the uncertainty is given with reference to the ERA5 footprint, which is achieved by filtering all data sets to match the ERA5 footprint (thick curves). For RO and RS92 the uncertainties based on un-smoothed data are also shown (thin curves). The found standard deviation estimates have been vertically smoothed with a 10 grid points box filter.

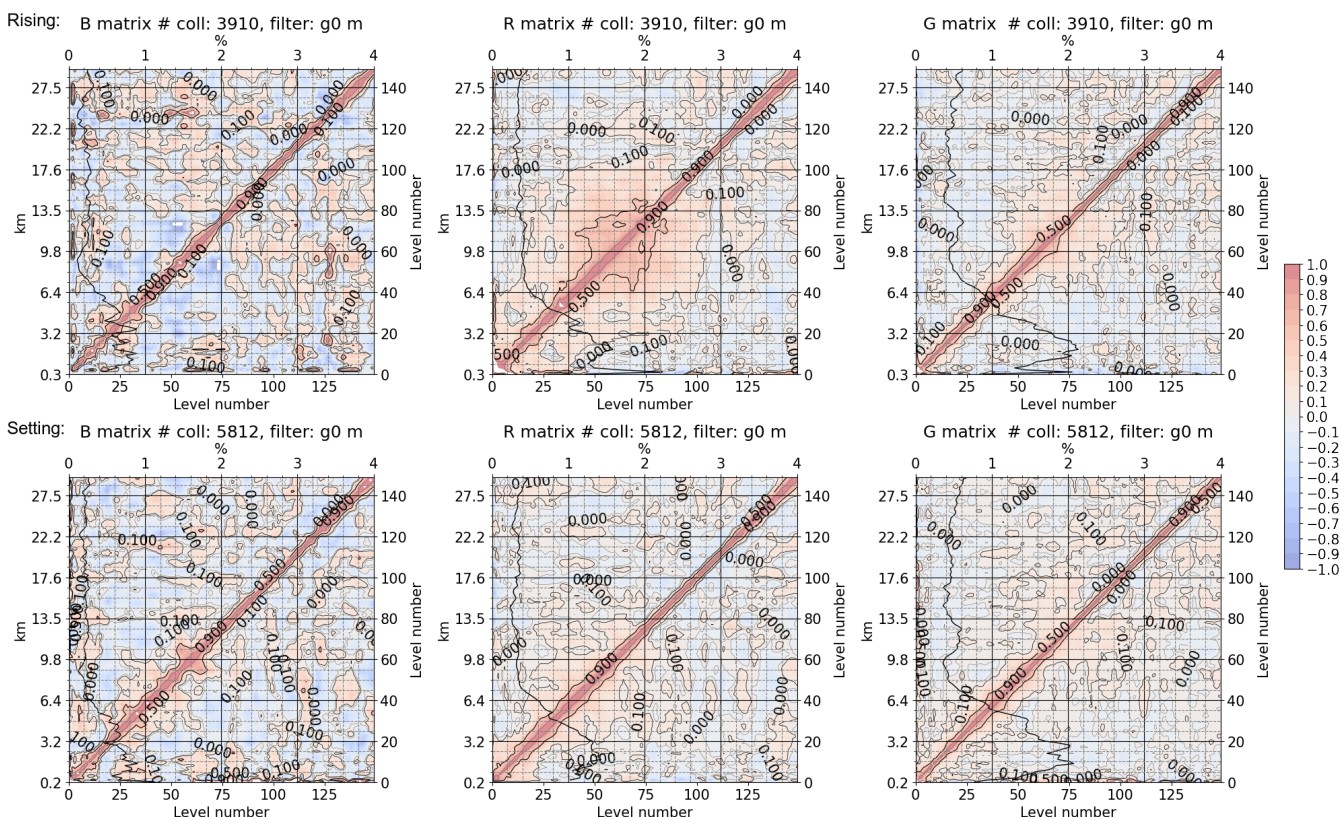

**Figure 11.** G3CH estimate of ERA5 (a,d), RO (b,e) and RS92 (c,f) refractivity vertical error covariance matrices at middle latitude. Rising occultations (a,b,c) and setting occultations (d,e,f). The covariance matrices are plotted as correlation matrices (diagonal = 1) with superimposed standard deviation as function of height (black line), plotted on the left and upper axes. The standard deviation is given in percent of the mean ERA5 forecast refractivity. The covariance matrices have been truncated at 30 km where the RS92 data sparseness starts to destabilize the results. 9722 collocated profile triplets were used for these covariance estimates. For these estimates no smoothing was applied on any of the data sets.



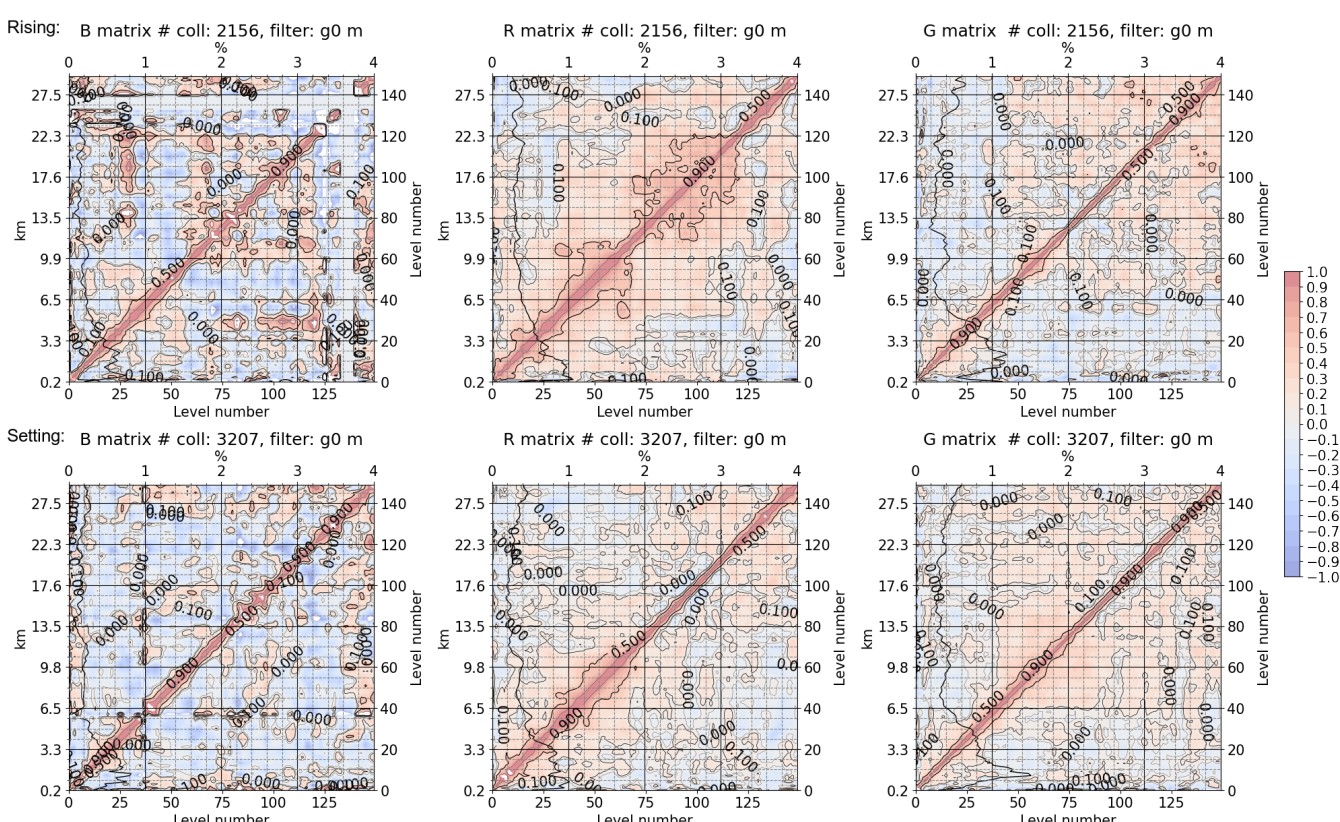

**Figure 12.** Same as in Fig. 11, but for high latitudes.



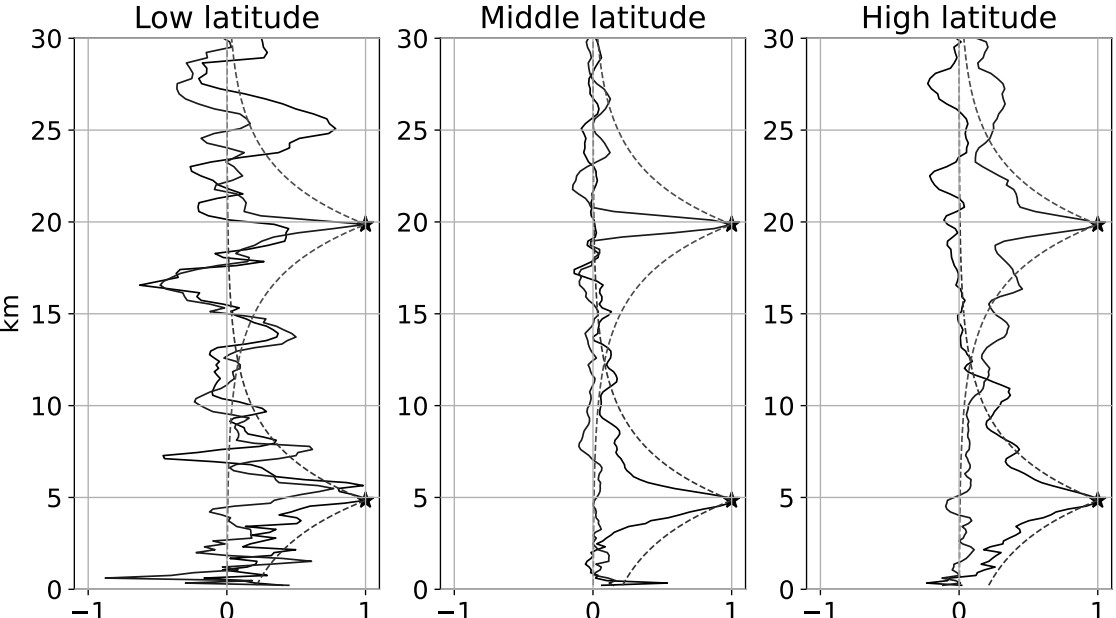

**Figure 13.** Estimate of RO vertical error correlations at approximately 5 and 20 km for low, middle and high latitudes (full curves). The dashed lines show exponential correlation with at three km decay length. No smoothing was applied on any of the data sets before the generalized G3CH was applied.