# Peer review of "Estimation of refractivity uncertainties and vertical error correlations in collocated radio occultations, radiosondes and model forecasts"

_Atmospheric Measurement Techniques, 2022_

## Referee Comment (RC2)

20 May 2022

**Review of:**
**"**Estimation of refractivity uncertainties and vertical error correlations in collocated radio occultations, radiosondes and model forecasts**",** by Nielsen et al., submitted to AMT. AMT-2022-121.  Version dated 9 May 2022.

**General comments**

1.      This paper presents a generalisation of the three-cornered hat (3CH) method for assessing the error statistics of 3 independent collocated data sets.  The generalisation (which is new) extends the method to error covariances of vector quantities (rather than error variances of scalar quantities).  The method is applied to the following data: refractivity profiles retrieved from radio occultation (RO) data, refractivity profiles calculated from radiosonde profiles (GRUAN RS92 data), and refractivity profiles calculated from ERA5 (ECMWF reanalysis) short-range forecasts.  The error covariance of each data set is derived using the 3CH method.  The sensitivities of the method to collocation criteria and to vertical smoothing are assessed.  The implications of the results for providing new and useful estimates of RO observation uncertainties are discussed.

2.      This is a very interesting paper, and the results should be useful to the RO and NWP data assimilation communities.  The results appear to represent a sound application of the method, and they are discussed thoughtfully.

3.      I have some concerns over aspects of the discussion, particularly with respect to the concept of the "true" profile and the discussion of error correlations between data sets.  However, addressing the issues in the text should not have any effect on the results.

4.      I therefore recommend that this paper could become suitable for publication subject to minor revision to address the detailed comments and editorial points below.

**Detailed comments**

5.      p.1-2, l.23-25: "This is likely because of the requirements — that the errors of the three data sets must be uncorrelated, and that the data sets must truly represent the same property with the same footprint in time and space — that are seldom met."  This is not the main reason; it is that NWP DA theory requires that the errors of representation (all of them) are considered as part of the observation error, not part of the NWP forecast error, even though they arise because of the NWP system's limited ability to represent the real world.  With the 3CH method it is not clear how the errors of representation will be portioned between the 3 data sets.  (The Desroziers method does not have this problem.)

6.      p.2, l.56-56: "The term vertical footprint of a data set is used here in the meaning width of an ideal physical refractivity feature, shaped as a delta function, mapped to the resolved representation of refractivity, for the given data set."  This is not very clear, partly because a (Dirac) delta function has zero width.  Please define more clearly what is meant to "vertical footprint", how it differs from "vertical resolution", and how it is related to the vertical smoothing discussed later.

7.      p.3, l.62-63: "t is the actual refractivity at a vertical line above the RO reference coordinates at the RO reference time".  3CH method, i.e. the solution of simultaneous equations described later in the paper, makes no assumption about exactly what the

reference profile is.  In fact, this is the paradox of the 3CH method, as discussed by O'Carroll et al. 2007 (appendix to their paper).  The solution to the paradox is through the appreciation that non-zero correlations of error (assumed to be zero in the 3CH method) arise because each of the data sets represents different spatial scales.  Alternatively, these can be considered as correlated errors of representation when the data are assumed to measure the same scales.  There are also the error correlations caused by space/time collocation differences.  In general, the paper discusses very well the issues of scales and error correlations, but it would be helpful to point out that they also are also related to the problem in defining the "true" profile.

8.     p.3, l.66: "we assume that".  This is a little confusing, because you say later that you actually processed the data to ensure that this was the case.  (You did not just assume that it was true.)

9.     p.3, l.68-69: "The $\varepsilon^R$ component represents the distortion of the underlying truth in a data set, as it is being mapped to the observation grid."  This is not clear.  To which observations does it apply?

10.    p.3, l.71-72: "representing the departure of the RO and RS92 trajectories in time and space from the vertical profile at the RO reference time."  Again, this is identifying the reference profile with the "true" profile, and the problems discussed in point 7 above apply.

11.    p.3, l.83: "intrinsic error".  Is this the same as epsilon, as introduced on line 61?

12.    p.3, l.88: "$\varepsilon^R$".  This is the representation error for this definition of the true profile, but it would not be the appropriate representation error for use in NWP DA.  See point 5 above.

13.    p.4, l.105-107: "model forecast …".  What is the range of this forecast, e.g. 3-, 6- or 9-hour forecasts interpolated to the RO observation time?
Also, "is model forecast" → "contains forecasts"?

14.    p.4, l.125: "we assume no cross correlation components".  Yes, this is what the algebra of the 3CH method assumes, but it is the weaknesses in this assumption that represent the problems with the method – see point 7 above.

15.    p.4, l.127: "it can without loss of generality be assumed that all three data sets are bias free".  Again, this is ambiguous – you did not just assume it – you actually ensured that it was true through the data processing (lines 151-152).

16.    p.6, l.167-168: "However, if two data sets have similar vertical footprints, differing from the vertical footprint of t, these two data sets will have cross-correlated errors".  Yes, they will have correlated errors, almost independently of how t is defined – see point 7 above.

17.    p.7, l.180-191: "Uncertainty estimates … . In the three cornered hat analysis, the data set with the largest footprint determines the common footprint to be used for all three data sets.".   I don't think this is true – see point 7 above.  It is certainly not consistent with the assumption that the "true" profile (relatively to which all errors are assessed) is a vertical profile at the nominal RO location.

Also, the use of the ERA5 scale as a common scale to which all data are smoothed is certainly a good strategy for the reasons stated but, again, it is not consistent with the definition of the true profile.

18.     p.8, l.205: "between 50 km and 300 km".  This must reduce the sample size by a factor of 36.  It raises the question of question the sample size is still big enough.  From the results it appears to be so, but this may be worth a comment.
Also, please could you comment on the related problem of temporal collocation window.

19.     p.8, l.213-214: "For each data set filtering has been performed, not on the data set itself, but on the two other complementing data sets (see figure legends)."  This is not very clear, and the figure caption is not any clearer.  I think I understand what has been done, but a few more words of explanation would be helpful, e.g. for the calculation for the curve shown for RO, only the RS92 and ERA5 data have been smoothed?

20.     p.9, l.253-254: "It is worth noticing that the estimated vertical correlations of RS92 are larger for setting than for rising RO at high latitudes, especially between 6 and 22 km. So the G3CH fails to give an independent estimate of the RS92 correlations."  This is a helpful warning; it is another illustration of a weakness in the 3CH and G3CH methods, through their implicit assumption that correlated errors (between data sets) are zero.

21.     p.10, l.269-273: "In the derivation of G3CH representativeness is defined with reference to given scales in space and time of the truth. The truth is assumed to have smaller footprint 270 than any of the involved data sets. We choose for all data sets to report the estimated uncertainty boundaries with reference to the estimated footprint of the ERA5 data set. … This operation is equivalent to define the truth t with reference to the ERA5 footprint if one will."  This is not consistent with 1.2 para 1.  It again illustrates that the choice of t is somewhat arbitrary with this method, and that the error correlations will change according to the scale of the "truth".

22.     p.10, l.287: "The increase of uncertainty in the troposphere is smaller at high latitude".  Why is this?  It may seem an obvious point, but you would expect larger % errors in the tropical lower troposphere because the absolute values of humidity are highest there and hence the collocation errors in refractivity will also be highest.

23.     p.11, l.331: "model" → "NWP"?

**Editorial comments**

24.     Throughout.  "data sets", "datasets" or "data-sets".  Consistency.

25.     Throughout.  "data" is usually used as plural, but in a few places as singular. Consistency.

26.     p.4, l.98: "has" → "have".  Also, l.103 and l.104.

27.     p.4, l.109: "spans" → "span".

28.     p.6, l.155: "is" → "are".

29.     p.8, l.224: "happens" → "happen".

30. p.10, l.274: "includes" → ", include"?

31. p.10, l.289: "does" → "do".

32. p.11, l.321: "promises" → "promise".

33. p.11, l.325: "forecast" → "forecasts".

34. p.22, Figure 7 caption: "easiest" → "most easily".

---

## Author Comment (AC1)

**Author response to Reviewer # 1, John Eyre: Estimation of refractivity uncertainties and vertical error correlations in collocated radio occultations, radiosondes and model forecasts**

Johannes K. Nielsen[1], Hans Gleisner[1], Stig Syndergaard[1], and Kent B. Lauritsen[1]

[1]Danish Meteorological Institute
[1]Lyngbyvej 100, DK 2100, Copenhagen, Denmark

**Correspondence:** Johannes K. Nielsen (jkn@dmi.dk)

**1   Authors response:**

The authors wish to thank for the review. We acknowledge the comments about the "truth" being in principle unknown in 3CH analysis, and we shall reformulate sentences that explicitly assumes a specific "truth". However we will maintain the view that, for heuristic reasons, it may be convenient to imagine an underlying true profile for each collocation set during the derivation or application of G3CH. Certainly one must not be mislead to conclude anything about the nature of that truth from 3CH. We are also grateful for the clarifying and editorial comments raised by the reviewer.

All the raised issues are addressed below:

**1.1   Detailed Comments:**

5. p.1-2, l.23-25: "This is likely because of the requirements — that the errors of the three data sets must be uncorrelated, and that the data sets must truly represent the same property with the same footprint in time and space — that are seldom met." This is not the main reason; it is that NWP DA theory requires that the errors of representation (all of them) are considered as part of the observation error, not part of the NWP forecast error, even though they arise because of the NWP system's limited ability to represent the real world. With the 3CH method it is not clear how the errors of representation will be portioned between the 3 data sets. (The Desroziers method does not have this problem.)

**Answer 5:** We have replaced the sentence with:
"This is likely because in NWP data assimilation all the model representativeness errors, including forward modeling errors, are considered as a part of the observation error. The 3CH method is not targeted specifically at NWP applications. This means that all three data sets involved are treated equally as a start, thus they are all assumed to contain representativeness errors,

with respect to the underlying truth. In order to use results obtained from the 3CH analysis it is necessary to consider, for each particular application, how representativeness errors are distributed among the involved data sets, and this is not always possible to find out."

6. p.2, l.56-56: "The term vertical footprint of a data set is used here in the meaning width of an ideal physical refractivity feature, shaped as a delta function, mapped to the resolved representation of refractivity, for the given data set." This is not very clear, partly because a (Dirac) delta function has zero width. Please define more clearly what is meant to "vertical footprint", how it differs from "vertical resolution", and how it is related to the vertical smoothing discussed later.

**Answer 6:** We have changed the mentioned sentence to this formulation (with a reference) "The term *vertical footprint* of a data set is used here in the same way as in Semane et al. (2022): *The vertical scale that an observation value represents.*"

7. p.3, l.62-63: "t is the actual refractivity at a vertical line above the RO reference coordinates at the RO reference time". 3CH method, i.e. the solution of simultaneous equations described later in the paper, makes no assumption about exactly what the reference profile is. In fact, this is the paradox of the 3CH method, as discussed by O'Carroll et al. 2007 (appendix to their paper). The solution to the paradox is through the appreciation that non-zero correlations of error (assumed to be zero in the 3CH method) arise because each of the data sets represents different spatial scales. Alternatively, these can be considered as correlated errors of representation when the data are assumed to measure the same scales. There are also the error correlations caused by space/time collocation differences. In general, the paper discusses very well the issues of scales and error correlations, but it would be helpful to point out that they also are also related to the problem in defining the "true" profile.

**Answer 7:** We have made this substitution at line 62: "In the context of this paper t is the actual refractivity at a vertical line above the RO reference coordinates at the RO reference time, defined with respect to given finite footprints in space and time, which may differ from the footprints of all three data sets."

— has been changed to:

"The G3CH does not make any assumptions about exactly what the true profile $\mathbf{t}$ is. $\mathbf{t}$ may be thought of as defined with respect to a given but unknown finite footprints in space and time, which may differ from the footprints of all three data sets."

Error cross correlations caused by space/time collocation differences are addressed through the discussion of $\varepsilon^C$ and $\varepsilon^G$ components in line 75 to 78. We assume that RS92 and RO sampling paths are uncorrelated, and that could in theory be disputed, but we anticipate that this issue is of minor importance. But one thing that could play a role, as also mentioned by Reviewer #3, would be horizontal cross correlations between ERA5 and RO, which are at least of similar horizontal scale. We have substituted:

"Hence the $\boldsymbol{\varepsilon}^G$ term contains no cross correlations, and consequently it will be correctly attributed to the RO and RS92 data by the G3CH procedure."

with:

"Hence the $\boldsymbol{\varepsilon}^G$ term can be assumed to contain no cross correlations. However, there are potentially error cross correlation

components arising from spatial correlations between the data sets, that we cannot assess. This could for example be the case for ERA5 and RO, because these are sampled on similar horizontal scales."

8: p.3, l.66: "we assume that". This is a little confusing, because you say later that you actually processed the data to ensure that this was the case. (You did not just assume that it was true.)

**Answer 8:** The sentence have been changed to "..., but for each subset of collocated triplets being analyzed, we remove systematic error differences between the three involved data subsets prior to the analysis."

9. p.3, l.68-69: "The $\varepsilon^R$ component represents the distortion of the underlying truth in a data set, as it is being mapped to the observation grid." This is not clear. To which observations does it apply?

**Answer 9:** In the "Error components" section error components are discussed generically such that the discussion applies for all 3 datasets. The model forecast is virtually treated as just another observation in this context. We have inserted a clarifying sentence at line 41: "The three data sets are treated on equal terms such that none of them are considered more or less representative for the truth a priori, thus the analysis will also provide estimates of ERA5 ... "

10. p.3, l.71-72: "representing the departure of the RO and RS92 trajectories in time and space from the vertical profile at the RO reference time." Again, this is identifying the reference profile with the "true" profile, and the problems discussed in point 7 above apply.

**Answer, 10:** Correct. We have changed the formulation to: "Especially $\varepsilon^R$ contains a geometric error component, $\varepsilon^G$, representing the departure of the ERA5, RO and RS92 vertical or skewed profiles in time and space from the unknown true profile at the RO reference coordinates."

And in line 74 we have added: "The used forward operator estimates refractivity along a 1 dimensional assumed vertical line, and this has an impact on the uncertainty estimates. Thus, the RO observation errors estimated by the 3CH method in this paper are applicable for variational assimilation with a 1D operator, but not for 2D/3D operators."

11. p.3, l.83: "intrinsic error". Is this the same as epsilon, as introduced on line 61?

**Answer 11:** Yes. "Intrinsic error " is maybe confusing. We have changed it to "observation error"

12. p.3, l.88: "$\varepsilon^R$". This is the representation error for this definition of the true profile, but it would not be the appropriate representation error for use in NWP DA. See point 5 above.

**Answer, 12:** We agree with this statement.

13. p.4, l.105-107: "model forecast ...". What is the range of this forecast, e.g. 3-, 6- or 9-hour forecasts interpolated to the RO observation time? Also, "is model forecast" → "contains forecasts"?

**Answer, 13:** The ERA5 forecasts have been downloaded in 3 hourly intervals. For each profile two fields of as little forecast time as possible are chosen, such that the RO and RS92 profiles are outside the analysis windows used for initialization of these forecasts. We have inserted this in line 107:

"Effectively this implies that the used verification times runs from 3 to 15 hours, and the ERA5 uncertainty is assumed to be constant in this time range."

We have changed "is model forecast" → "contains forecasts"?

14. p.4, l.125: "we assume no cross correlation components". Yes, this is what the algebra of the 3CH method assumes, but it is the weaknesses in this assumption that represent the problems with the method – see point 7 above.
**Answer 14:** That has been noted.

15. p.4, l.127: "it can without loss of generality be assumed that all three data sets are bias free". Again, this is ambiguous – you did not just assume it – you actually ensured that it was true through the data processing (lines 151-152).
**Answer 15:** We have changed the formulation accordingly: "In the analysis in the present paper we only estimate the random uncertainties, so it can without loss of generality be assumed that all three data sets are bias free. This can in practice be ensured by subtracting the mean of each data set prior to the analysis." has been changed to: "In the present paper we only estimate the random uncertainties. In practice we remove biases in each subset of collocations where G3CH is to be applied by subtracting the subset mean of each of the three data sets prior to the analysis. So in the following derivation we can assume that all data are bias free."

16. p.6, l.167-168: "However, if two data sets have similar vertical footprints, differing from the vertical footprint of t, these two data sets will have cross-correlated errors". Yes, they will have correlated errors, almost independently of how t is defined – see point 7 above.
**Answer 16:** We agree on that, no action needed.

17. p.7, l.180-191: "Uncertainty estimates ... . In the three cornered hat analysis, the data set with the largest footprint determines the common footprint to be used for all three data sets.". I don't think this is true – see point 7 above. It is certainly not consistent with the assumption that the "true" profile (relatively to which all errors are assessed) is a vertical profile at the nominal RO location. Also, the use of the ERA5 scale as a common scale to which all data are smoothed is certainly a good strategy for the reasons stated but, again, it is not consistent with the definition of the true profile.
**Answer 17:** Right, the formulation is wrong. We have changed this: "In this analysis.." —> "In our approach..."

18. p.8, l.205: "between 50 km and 300 km". This must reduce the sample size by a factor of 36. It raises the question of question the sample size is still big enough. From the results it appears to be so, but this may be worth a comment. Also, please could you comment on the related problem of temporal collocation window.
**Answer 18:** The covariance estimates near 50 km collocation distance are indeed uncertain, as is also seen in Fig 5. We believe this is the main reason for the noise in the uncertainty profiles. We tried at an early stage to include both distance in time and space in the analysis, and found that the impact of time distance between GRUAN and RO profiles on the GRUAN - RO temperature standard deviation was negligible, at least as long as the time criterion was less than 3 hours, so we left that

out of the analysis.

115  The text has been changed to this: "We are calculating the G3CH estimates of covariance matrices for a sequence of collocation criteria (between 50 km and 300 km) and use these to extrapolate all covariance matrices to 0 km collocation distance, with a linear fit to the full covariance matrices as function of the squared collocation distance. The effect of varying the temporal collocation window is small, so we have excluded that from the analysis."

19. p.8, l.213-214: "For each data set filtering has been performed, not on the data set itself, but on the two other complementing
120  data sets (see figure legends)." This is not very clear, and the figure caption is not any clearer. I think I understand what has been done, but a few more words of explanation would be helpful, e.g. for the calculation for the curve shown for RO, only the RS92 and ERA5 data have been smoothed?

**Answer 19:** That is correct. It is as explained, but maybe it would help with more motivation or explanation of why this procedure is chosen. We have inserted:

125  "The idea is basically to probe the vertical footprint of one data set with two other data sets of varying footprint." On line 214.

20. p.9, l.253-254: "It is worth noticing that the estimated vertical correlations of RS92 are larger for setting than for rising RO at high latitudes, especially between 6 and 22 km. So the G3CH fails to give an independent estimate of the RS92 correlations." This is a helpful warning; it is another illustration of a weakness in the 3CH and G3CH methods, through their implicit assumption that correlated errors (between data sets) are zero.

130  **Answer 20:** We agree that the explanation can be error cross correlations.

21. p.10, l.269-273: "In the derivation of G3CH representativeness is defined with reference to given scales in space and time of the truth. The truth is assumed to have smaller footprint 270 than any of the involved data sets. We choose for all data sets to report the estimated uncertainty boundaries with reference to the estimated footprint of the ERA5 data set. ... This operation is equivalent to define the truth t with reference to the ERA5 footprint if one will." This is not consistent with 1.2 para 1. It again
135  illustrates that the choice of t is somewhat arbitrary with this method, and that the error correlations will change according to the scale of the "truth".

**Answer 21:** We agree that 1.2 para 1 is misleading because it suggests that the G3CH scheme works for a specific truth, and we have taken that part out of the paper, and we have removed this sentence at line 270: "The truth is assumed to have smaller footprint than any of the involved data sets." And even though it is not really wrong, we have removed this sentence at line
140  272: "This operation is equivalent to define the truth t with reference to the ERA5 footprint if one will."

22. p.10, l.287: "The increase of uncertainty in the troposphere is smaller at high latitude". Why is this? It may seem an obvious point, but you would expect larger errors in the tropical lower troposphere because the absolute values of humidity are highest there and hence the collocation errors in refractivity will also be highest.

**Answer 22:** We agree that tropospheric humidity is causing a lot of uncertainty in the tropical troposphere, but since the paper
145  is not really addressing physical interpretations, we think that a comment on this would appear unmotivated in this context.

23. p.11, l.331: "model" → "NWP"?

**Answer 23:** Changed.

Editorial comments 24. Throughout. "data sets", "datasets" or "data-sets". Consistency.

**Answer 24:** Changed to "data sets"

25. Throughout. "data" is usually used as plural, but in a few places as singular. Consistency.

**Answer 25:** Thanks.

26. p.4, l.98: "has" → "have". Also, l.103 and l.104.

**Answer 26:** We have chosen singular

27. p.4, l.109: "spans" → "span".

**Answer 27:** We have chosen singular

28. p.6, l.155: "is" → "are".

**Answer 28:** Not found

29. p.8, l.224: "happens" → "happen".

**Answer 29:** Changed

30. p.10, l.274: "includes" → ", include"?

**Answer: 30:** Changed

31. p.10, l.289: "does" → "do".

**Answer: 31:** Changed

32. p.11, l.321: "promises" → "promise".

**Answer: 32:** Changed

33. p.11, l.325: "forecast" → "forecasts".

**Answer: 33:** Changed

34. p.22, Figure 7 caption: "easiest" → "most easily"

**Answer: 34:** Changed

**170 References**

Semane, N., Anthes, R., Sjoberg, J., Healy, S., and Ruston, B.: Comparison of Desroziers and Three-Cornered Hat Methods for Estimating COSMIC-2 Bending Angle Uncertainties, Journal of Atmospheric and Oceanic Technology, 39, 929–939, https://doi.org/10.1175/JTECH-D-21-0175.1, 2022.

---

## Author Comment (AC2)

**Author response to Reviewer # 2, Paul Poli: Estimation of refractivity uncertainties and vertical error correlations in collocated radio occultations, radiosondes and model forecasts**

Johannes K. Nielsen[1], Hans Gleisner[1], Stig Syndergaard[1], and Kent B. Lauritsen[1]

[1]Danish Meteorological Institute
[1]Lyngbyvej 100, DK 2100, Copenhagen, Denmark

**Correspondence:** Johannes K. Nielsen (jkn@dmi.dk)

**1 Authors response:**

We acknowledge the constructive suggestions from Reviewer #2, which have helped to clarify several aspects in the manuscript. We have considered all suggestions, and adopted most of them. All issues raised by Reviewer # 2 are addressed below, and we have also inserted some requested figures.

**2 Detailed comments:**

(1) Given that the authors make an explicit attempt to tie the terminology to established documents like the GUM, other prior relevant publications may deserve to be cited, namely those that already considered the GUM and its applicability to Earth Observation data, e.g., from the FIDUCEO project: Merchant, C. , G. Holl, J. P. D. Mittaz, and E. R. Woolliams. 2019: Radiance Uncertainty Characterisation to Facilitate Climate Data Record Creation. Remote Sensing 11, no. 5: 474. doi:10.3390/rs11050474

**Answer 1:** Thank you for suggesting this reference. We have inserted a citation.

(2) Section 1.2 mentions "RO reference coordinates" and "RO reference time": how are they defined in the present work?

**Answer 2:** We have inserted this in Sect. 1.2: "The RO reference coordinate is the point at which a straight line between the GNSS satellite and the receiving Low Earth Orbiter tangents the Earth ellipsoid."

(3) Is it possible to indicate (or cite the appropriate reference) for the step where radiosondes and ERA5 data are projected into refractivity space?

**Answer 3:** We have inserted this (see also Answer 7): "The refractivity calculation is done with the method described in the

ROPP user guide: https://www.romsaf.org/romsaf_ropp_ug_fm.pdf." The GRAUN data are mapped to the same 137 level grid as ERA5 before being forward modelled to refractivity space.

20   (4) The approach to calculate epsilon C and epsilon X needs to be detailed, and preferably with dedicated equations for clarity. This would also remove the need for forward references to sections 4.2 and 4.3 in section 1.2.

**Answer 4:** Instead of referring to Sections 4.2 and 4.3 we have added this:

"We are able to remove $\varepsilon^C$ and the $\varepsilon^X$ components of the three data sets, by adding the following additional analysis steps to the G3CH. The $\varepsilon^C$ covariance matrix, $\mathbf{C}^C$, is eliminated by first calculating G3CH estimated covariance matrices $\mathbf{C}_i$ for a

25   series of collocation subsets, sampled from areas of decreasing size around th RO reference coordinates. Next, the sequence of decreasing covariance estimates is extrapolated to the virtual zero-area case $\mathbf{C}_0$. $\mathbf{C}_i^C = \mathbf{C}_i - \mathbf{C}_0$. Subsequently the $\varepsilon^X$ covariance matrix, $\mathbf{C}^X$, is eliminated by smooting all three data sets such that they have the same vertical footprint, and then calculate for each data set a covaraince matrix $\mathbf{C}_s$ with G3CH from the smoothed data sets. $\mathbf{C}^X = \mathbf{C}_0 - \mathbf{C}_s$. So the observation error covariance matrices that we estimate in the end includes only measurement error $\varepsilon^I$ and representativeness error $\varepsilon^R$."

30   (5) A map showing the locations (or a density map) of the 15,997 selected collocations may be a useful information for the readers.

**Answer 5:** We will add this map as supplementary material.

[Figure]

**Figure 1.** Reference locations of GRUAN RS92 sondes.

(6) Is it possible to comment on the possibility that the two steps of cubic spline interpolation, and of refractivity computations, both applied to ERA5 and radiosonde data, may (each) introduce correlations of uncertainties between these two datasets? Similarly, given that the assimilation of RO data improves the quality of a reanalysis, what are the prospects for some structural correlation between ERA5 forecast (even if, not analyses) and RO data?

**Answer 6.1:** The spline interpolation is done on a length scale which is smaller than the common footprint from ERA5 and therefore we do not expect it to create correlations on the length scales where we draw conclusions here (> 300 m)

**Answer 6.2:** The forward model does introduce some correlation through interpolation from the model grid to the observation grid. Again, these correlations will appear on length scales smaller than the common footprint, and as such they should not influence the estimated refractivity uncertainty.

**Answer 6.3:** Such contamination would manifest itself as a bias change in the analysis which subsequently spills over to the forecast. Given that biases are eliminated in the application of G3CH, it can be assumed that no such information spill over is taking place. (No changes made to the manuscript as response to (6))

(7) The numbers of vertical levels in each dataset may need to be introduced in the data section.

**Answer 7:** We have inserted this information:

Line 95: "Metop and COSMIC-1 missions (Gleisner et al., 2020), interpolated to 247 levels."

And on line 110: "The RS92 temperature, humidity and pressure variables have been interpolated with cubic splines to the 137 ERA5 model levels, hereafter the ERA5 and RS92 variables have been forward modeled to refractivity at the RO vertical grid of 247 levels(Lewis, 2009). The refractivity calculation is done with the method described in the ROPP user guide: https://www.romsaf.org/romsaf_ropp_ug_fm.pdf."

(8) As the work is using model forecast, whose quality decays as the integration time increases, may one expect a sensitivity of the results to the forecast integration time?

**Answer 8:** That is correct, we have chosen to ignore that, so the ERA5 uncertainty is representative for 3 to 15 hours forecasts. We have inserted this in line 107: "Effectively this implies that the used verification times runs from 3 to 15 hours, and the ERA5 uncertainty is assumed to be constant in this time range."

(9) Relying on correlations to pick-up a signal exposes one to be sensitive to any transient or structural correlation that may exist in the input data that are correlated, whatever the reason (true signal, artefact of pre-processing, matching bias, …). In the present case, the step of bias removal seems to be limited to a mean profile subtraction, carried out at the scale of each entire dataset (RO, RS, ERA5), is this correct? If so, this would leave, present in the data, all the bias(es) that may exist within each subset of analysis. Would it be possible to consider applying the bias removal in each subset (i.e. at the step when expectation values are computed, modifying slightly equation (7) to introduce the removal of the means), and then display (or report on) how much this changes the results (or not).

**Answer 9:** The biases are removed for each subset separately in the application of G3CH, such that for instance when looking at rising occultations at middle lattitude, the 3 bias vectors are calculated individually for this subset before applying 3CH. We

can see how this is not clear from the paper, and we have changed the formulations:

Line 66: "...but for each subset of collocated triplets being analyzed, we remove systematic error differences between the three involved data subsets prior to the analysis."

Line 126:"In the present paper we only estimate the random uncertainties. In practice we remove biases in each subset of collocations where G3CH is to be applied by subtracting the subset mean of each of the three data sets prior to the analysis. So in the following derivation we can assume that all data are bias free."

(10) Do the brackets in the right-hand side of in Equation (7) reflect the actual implementation? (i.e. averages are computed after adding all cross-products?)

**Answer 10:** No, each cross-product is averaged individually before adding the 3 terms.

(11) Can you clarify how the data subsets (the sub-spaces in which expectation values and hence correlations are computed) are defined? This seems to be, at least initially, based solely by considering the vertical dimension, but then later in the paper other dimensions (for computing the correlations and presenting the results) are introduced. This may be done with various sets of subscripts (for the various dimensions: vertical, latitude band, . . . ). In the ideal case where one would have many events, one could consider to compute these error estimates with subsets defined spatially (e.g. 5 deg x 5 deg). The resulting geographic patterns that may be obtained could be of interest.

**Answer 11:** The choice of the GRUAN RS92 profiles as demonstration data set is the limitation here. While being a limited quality checked data set of high accuracy, it only covers a very sporadic area. As it is also evident from the results, especially in the tropics, the data set is barely large enough to justify separation into 3 latitude bands. In future applications one might (probably should) choose a much larger set of sondes with better coverage.

(12) Would it be possible to define early on, i.e., in the methodology section, the 'raw uncertainty estimates' mentioned in the results section? Also, the sigma symbol may deserve to be introduced numerically with an equation.

**Answer 12:** Good question: "Raw uncertainty estimates" are uncertainty estimates calculated from some subset of triple collocations without correcting for collocation error and without performing any smoothing. Unfortunately we have used the term "raw" in Fig. 10, refering to estmates that have been collocation corrected but still unfiltered (this will be corrected). We insert this in the Figure 9 caption: "and e.g., $\frac{\sigma_{ERA5}}{\sigma_{ERA5,0}}$ in subplot **(a)** means uncertainty estimate of ERA5, given filtering of the data set mentioned in the legend — in this case RS92, divided with the uncertainty estimate of ERA5, obtained without filtering."

(13) In figure 7, does the "0 m" line refer to no filter? If so, such a filter has an infinitely small width (Dirac), but is probably non-zero.

**Answer 13:** Yes "0 m" means zero width and is equivalent to no filtering; multiplication with unit matrix.

(14) The results in figure 7 indicate that as one filters out small-scale variability in both other datasets, the dataset that appears to be most affected in its 'error' estimate is ERA5 (this one presents the largest spread, nearly 1filter, in the lower troposphere).

This would be consistent with that dataset containing the least small-scale vertical information, given that equation (7) suggests that for an error estimate to increase, there are two pathways: the cross-products of differences with respect to the two other datasets increase (first two terms in the sum), and/or the cross-product of the differences between the two other datasets decreases (third term, negative sign). The latter may be the mechanism by which removing small-scale information in RS and RO data (thus reducing the differences RO-RS) leads to ERA5 to appear of worse quality (when in fact its quality should be independent of that, but here the method uses the other data as references). Such hand-waving comments (for lack of a better expression) may be tried with a simple toy model. Similarly, the dataset whose 'error' estimate is the least affected by filtering the small-scale variability in the two other datasets seems to be the radiosondes, which is also consistent with that data source possibly containing the most small-scale vertical information in the lower troposphere (or is the figure 7(c), truncated at a maximum of 2.0

**Answer 14:** We agree with the interpretation, and we only want to clarify that ERA5 estimated uncertainty should be independent on the quality of RO and RS: This is in principle true, but it fails to hold when the vertical footprints of the RO and RS data are smaller than that of ERA5. We apologize for the truncation of Fig. 7 (c), which was chosen to make the relative small differences higher up visible. We will exchange it with a version with expanded x axis.

(15) Figure 9 shows seemingly slightly different results because in this case one considers smoothing on only one (other) dataset a time. However, one finds consistency. When (only) RS or (only) RO data are filtered (in (a) and (c), respectively), then either one of the two may start to resemble more to ERA5 (but less to the other, i.e. RO or RS, respectively), so, in equation (7), the three terms that make up the total error estimate sees changes of different signs in its components (respectively, for the 3 terms in the right-hand-side of equation (7): decrease of differences ERA5-RS, no change ERA5-RO, and minus an increase of differences RS-RO – the net result is then a decrease of ERA5 estimated 'errors' when only RS is filtered). Similarly, this would explain that the 'error' estimate of RO increases in (b) when (only) RS data are filtered, making them resemble more ERA5 (respectively: increase of differences RO-RS, unchanged differences RO-ERA5, and minus a decrease of differences RS-ERA5). I note in passing that one missing piece of this puzzle would be to show what is happening to the error estimates of each dataset, when one filters that dataset only, and none of the other two datasets, as the results are not entirely predictable because they involve the sum of two terms moving in opposite directions, e.g., for RO error estimates, if filtering RO data: the differences RO-RS may increase, the differences RO-ERA5 may decrease, the differences ERA5-RS would be unchanged (so the net result is hard to predict – but such a thought experiment may help shed light on the optimal 'footprint' to characterize each dataset).

**Answer 15:** We show plots, where only the shown data set itself is filtered here:

[Figure]

**Figure 2.** G3CH uncertainties when only the variable at question is filtered.

As expected ERA5 gets further away from the other data when filtering is applied.

(16) With respect to the expected long-range correlations expected in RS data (but not picked-up as well as expected by the method), this may also be due to the choice of sub-setting considered here, analyzing together day-time and night-time data. The effects of the radiative corrections (leading to consistently positive or consistently negative differences in each profile) may, if not cancel out, possibly be reduced, when considered together. However, redoing the exercise by separating clearly night- and day- ascents (and possibly leaving aside those profiles 'in between'), may show slightly different results. Such a separation for the RS data would somehow echo the efforts made to separate between rising and setting events for the RO part. **Answer 16:** Thank you for this suggestion. We think it is interesting, but since the main focus here is to understand RO uncertainty, we will leave this out, and possibly try it in future applications where GRUAN data are more in focus.

(17) Figures 11-12, I fail to see the labels (a) to (f) (either add these or amend the figure caption?).
**Answer 17:** Thank you for spotting this, we shall fix that.

**References**

Gleisner, H., Lauritsen, K. B., Nielsen, J. K., and Syndergaard, S.: Evaluation of the 15-Year ROM SAF Monthly Mean GPS Radio Occultation Climate Data Record, Atmospheric Measurement Techniques, 13, 3081–3098, https://doi.org/10.5194/amt-13-3081-2020, 2020.

Lewis, H.: GRAS SAF Report 08 ROPP Thinner Algorithm, Tech. Rep. Ref: SAF/GRAS/METO/REP/GSR/008, EUMETSAT, 2009.

---

## Author Comment (AC3)

**Author response to Reviewer # 3, Anonymous: Estimation of refractivity uncertainties and vertical error correlations in collocated radio occultations, radiosondes and model forecasts**

Johannes K. Nielsen1, Hans Gleisner1, Stig Syndergaard1, and Kent B. Lauritsen1

1Danish Meteorological Institute 1Lyngbyvej 100, DK 2100, Copenhagen, Denmark **Correspondence:** Johannes K. Nielsen (jkn@dmi.dk)

**1** Authors response:**

5

We acknowledge the constructive suggestions from Reviewer #3, which has lead to substantial changes of the manuscript. Reviewer # 3 calls for clarification of assumptions and certain aspects of the argumentation, and indeed the questions raised has pointed out parts of the text that has lead to misunderstandings. We apologize for the lack of clarity, especially regarding the definition of "vertical footprint", which has understandably frustrated the reviewer. By addressing every question raised by

Reviewer # 3 below, and changing many formulations in the manuscript we hope that we have sufficiently closed all outstanding issues. Reviewer #3 calls for figures to illustrate some concepts of the paper. We cannot accomplish that, but we think that the explanations below and the revised manuscript eliminates the need for drawings.

**1.1 Detailed Comments:**

10 Line 36: this sentence is imprecise. One of the purposes of the paper is to take into account error correlations. Errors contain a random component. Therefore stating that the random error components are independent appears mis-leading. Random errors can be dependent and correlated. This should be re-phrased.

Answer, line 36: Right, that is unclear. The intention is to state that we anticipate error cross correlations, as defined in the paper, to be negligible. We have changed this formulation: "We apply the generalized 3CH (G3CH) to three data sets where the

15 random error components can be assumed to be truly independent." into: "We apply the generalized 3CH (G3CH) to three data sets where the random errors components can be assumed not to be interdependent, meaning that their error cross correlations are assumed to be negligible." Line 39: while it is true that the authors focus on vertical error correlation, they have not made a convincing case that error correlation might not arise for other reasons. It seems that the current analysis could proceed at a particular vertical level, in

- 20 which case it would seem incorrect to assume that all error correlation is from the vertical dimension. More on this point later. **Answer, line 39:** It is correct that the analysis can be performed on a particular vertical level. That would be the classical three cornered hat analysis, which has been studied several times in literature. One would obtain the diagonal elements of the covariance matrices, a perfectly valid analysis. We would like to note that the focus, in this paper, on vertical correlations does not imply that we assume that all errors are driven by vertical coupling. Likewise, in classical 3CH, it is not assumed that that
- 25 there are no vertical correlations, they are just not analyzed. Reviewer # 3 is right in that there can be error cross correlations between the data sets that we are not aware of. We have chosen 3 data sources, obtained with completely different techniques, as different as they can possibly be within available data sets. On this background assume that the cross correlations are negligible. However, we should add that the mechanisms that produce vertical correlations in refractivity profiles might include processes that depend on horizontal coupling or horizontal
- 30 representation. For instance, ERA5 and RO are both sampled over a horizontal range of the order of 100 km, which may imply that some errors arising from horizontal coupling could be cross correlated in the two data sets. We have included this point in the paper on line 76, where we have substituted: "Hence the  $\varepsilon^G$  term contains no cross correlations, and consequently it will be correctly attributed to the RO and RS92 data by the G3CH procedure." with:
- 35 "Hence the  $\varepsilon^G$  term can be assumed to contain no cross correlations. However, there are potentially error cross correlation components arising from spatial correlations between the data sets, that we cannot assess. This could for example be the case for ERA5 and RO, because these are sampled on similar horizontal scales."

Line 41: in light of the earlier sentences in this paragraph, are we to assume that the ERA5 and RS92 error covariance matrices contain off-diagonal terms only because of vertical error correlation? Please clarify.

- 40 Answer, line 41: No, as it is also stated above, we make no assumptions about the origin of the vertical error correlations. The G3CH approach presented does in fact allow correlations in any spatial or temporal dimension to be examined, but we only examine vertical error correlations. It is a perfectly valid and mathematical sound approach which is being used widely in NWP and satellite retrieval: For an atmospheric profile one can define the covariance of errors ei, ej of some property at two altitude levels zi and zj, covij =< eiej >. Horizontal error correlations may exist for instance between two ERA5 profiles at different locations, but that does not prevent one from studying the vertical error correlations separately.
- 45° unifient locations, but that does not prevent one from studying the vertical error correlations separate

Line 58: this sentence is not understood. Vertical footprint for RO profiles will be of order 10 km, which is much less than the distance between RO measurements for the data sets considered here.

Answer, line 58: Reviewer # 3 refers to this sentence: "The vertical footprint will typically be larger than the distance between data points." We see how the sentence can be misunderstood, and we have rephrased it to: "The vertical footprint will typically

50 be larger than the distance between the vertical height levels which the data values refer to." We remark that the vertical footprint of RO profiles is found in the paper to be of the order of a few hundred meters.

Line 63: we suggest the authors add a figure to the paper that defines precisely what is meant by "footprint" for the data sets and for truth. This same figure should clarify the term "observation grid" (Line 69), since the observations are available at random places and times, and not on a grid.

55 Answer, line 63: We realize that the term "vertical footprint" may stir some confusion here, and this issue was also raised by Reviewer # 1. We have tried, on line 56, to be more concise in the definition of this term with this formulation (and reference): "The term *vertical footprint* of a data set is used here in the same way as in Semane et al. (2022): *The vertical scale that an observation value represents.*"

Answer, line 69: The word "grid" is used here in the meaning of vertical levels on which the refractivity is expressed. It has

60 nothing to do with horizontal representation. We have changed line 69: " ... as it is being mapped to the vertical observation grid."

Line 66: assuming that systematic errors are removed, i.e. errors have no bias, is a confusing aspect of this paper. While it is true that the authors remove global means from the data sets, this does not imply the errors as analyzed contain no bias. The reason is that the authors use data subsets in the analysis (e.g. latitude subsets, collocation-distance subsets, etc.) and these

65 subsets may contain bias. An example would be lack of global bias arising because there are equal and opposite biases in the northern and southern hemispheres. The authors need to consider the possibility of biases in subsets of the data. If they take this into account, the analysis can proceed apace.

Answer, line 66: We have not been clear about this, so it is valid critique. As we explain in the response to Reviewer # 2, answer (9), the biases are removed for each subset separately in the application of G3CH, such that for instance when looking

70 at rising occultations at middle lattitude, the 3 bias vectors are calculated individually for this subset before applying 3CH. We can see how this is not clear from the paper, and we have changed the formulations: Line 66: "...but for each subset of collocated triplets being analyzed, we remove systematic error differences between the three involved data subsets prior to the analysis."

And in line 126: "Systematic differences between data subsets are in practice removed by subtracting the mean of each data

75 subset prior to the analysis."

But still we pool the Northern and Southern Hemispheres, and also for instance land and ocean, so within each subset there may be groups of data that contain different observation biases. So indeed, if one data set contains separate bias regimes, this will lead to an overestimate of random uncertainty for that variable. We cannot do anything about that with the data at hand, except for requiring larger data sets.

80 Line 69: we again recommend a figure be used to carefully define how the truth data set is "distorted" when mapped to the observation grid, and to carefully define what is meant by "representativeness error".

Answer, line 69: We cannot provide a drawing, but we can explain: The word "distortion" is chosen to include multiple sources

of representativeness error, all contributing to the inability of the observation to represent the truth. Here the definition of representativeness error differs from the NWP definition, which — as also pointed out by Reviewer #1 — attributes the model

- 85 representativeness errors and the forward model errors to the observation representativeness error. The representativeness errors are composed of several components, including limited resolution, spatial and temporal differences in sampling, processing errors, interpolation errors and errors caused by smoothing. They come about in very different ways in the three data sets. For instance, by far most of the RO observation error comes from representing the radio occultation as a 1 dimensional profile. In the processing a crucial assumption of spherical symmetry neglects horizontal gradients in the atmosphere. This component is
- 90 regarded as a representation error.

110

Line 83: based on the writing so far, "error cross correlations" are error correlations between data sets. It is not immediately obvious to this reviewer how finite footprints of the data sets would lead to such correlations. (Again, the suggested figure might help here). For example, if the data sets are not overlapping in space, why would finite footprints lead to error correlation? Also, please clarify whether the footprints alluded to are horizontal, vertical or both. This question is posed because in several hear of the space o

95 locations of the paper it is implied that vertical correlation is what leads to non-zero off-diagonal covariance, so vertical footprint would be relevant.

Answer, line 83: It is instructive to think of the following thought experiment: Consider two identical vertical profiles  $\{x_i, y_i\}$ , with good resolution, that is with small vertical footprint, and with no errors. For simplicity, one may think of the truth as a profile with the same resolution. That is, we have two profiles identical with the truth  $t_i$ . Suppose that we now smooth

- 100 these two profiles with the same filter. The smoothing introduces an error,  $\epsilon_i$ , which is identical for the two profiles  $\epsilon_i = x_{\text{smoothed},i} t_i = y_{\text{smoothed},i} t_i$ . Now consider a set of such paired profiles  $\{x_i, y_i\}_{i=1..N}$  sampled across the globe, that is we have two extremely well collocated data sets  $\{x_i\}_{i=1..N}$  and  $\{y_i\}_{i=1..N}$ . Let these two data sets be smoothed with the same filter, such that they have similar vertical footprints. The smoothing will cause both systematic and random error. Let us assume for simplicity that the systematic error is zero such that  $1/N \sum_i \epsilon_i = 0$ . When we calculate the error cross correlation
- 105 by averaging over the set of pairs we still get a non-vanishing error cross covariance  $cov(x, y) = 1/N \sum_{i} (x_{smoothed,i} t_i)(y_{smoothed,i} t_i) = 1/N \sum_{i} \epsilon_i \epsilon_i$ . We hope that this extreme case can help to illustrate how similar footprints can cause error cross correlations between two data sets.

We can see how the distinction between vertical and horizontal footprint is assumed to be implicit from the context some in the paper, and as such not stated clear enough. We have made sure that "*vertical footprint*" is stated explicitly throughout the paper.

Line 120: Please clarify the notation. Do the different epsilon terms (x,y,z) each decompose into components I, R, C, X in equation (1)? We assume that the vector here represents different values along the vertical dimension. That could be stated explicitly.

Answer, line 120: Yes, Sec. 1.2, describes the error components of all three data sets. The formulation "For a given refractivity

115 data profile, x,..." is not clear. We will change this to "For a given refractivity data profile from either of the three data sets RS92, RO, or EAR5 ..."

Lines 127-128: we have remarked earlier how the bias free assumption may not apply. Can the authors verify they have removed bias from all data subsets they have worked on? Subtraction of one global bias will not guarantee there are no biases in subsets of the data. Also, the statement that randomness implies "bias free" is mis-leading. Please modify this statement.

120 **Answer, lines 127-128:** True, it has not been stated clear enough that we remove biases for each subset individually. So this sentence:

"In the analysis in the present paper we only estimate the random uncertainties, so it can without loss of generality be assumed that all three data sets are bias free. This can in practice be ensured by subtracting the mean of each data set prior to the analysis."

125 has been changed to:

145

"In the present paper we only estimate the random uncertainties. In practice we remove biases in each subset of collocations where G3CH is to be applied by subtracting the subset mean of each of the three data sets prior to the analysis. So in the following derivation we can assume that all data are bias free."

Line 156: we raise again the concern that all data subsets might not be bias free. Has this been confirmed in the analysis?
Answer, line 156: As also mentioned above, each subset has been centered such the its mean is zero prior to G3CH analysis.

Line 167: we ask the authors to define the "vertical footprint of truth". The figure asked for earlier would again help here. .... Answer, line 167: For heuristic reasons we think it is convenient to think of the truth as referring to a certain vertical scale. However, to avoid further discussion we have removed this phrase: ".... differing from the vertical footprint of t"

Line 167: why wouldn't similarity of horizontal footprints also lead to cross-correlated errors?

- 135 Answer, line 167: That is a good point. As also mentioned in "Answer, line 39", there can be a cross-correlation arising from similarity between e.g. ERA5 and RO in horizontal footprint. We have a method to assess the vertical footprint and eliminate cross-correlations arising from similarity in vertical footprint, but we have no way to access the error cross correlation arising from similarity in horizontal footprint. This is a limitation of the method, and we have extended the formulation:
- "... if two data sets have similar vertical footprints, or if they are sampled at similar horizontal scales, these two data sets may
  have cross-correlated errors, and possibly biases. All biases are removed prior to application of G3CH, but the error cross correlations introduced by finite vertical footprints or similar horizontal scale may influence the result of G3CH.
  We also added a this statement at line 78:

"Hence the  $\varepsilon^{G}$  term can be assumed to contain no cross correlations. However, there are potentially error cross correlation components arising from spatial correlations between the data sets, that we cannot assess. This could for example be the case for ERA5 and RO, because these are sampled on similar horizontal scales."

Line 171: what is meant by "common grid"? What are the spacings of this grid?

**Answer, line 171:** The common vertical grid is 247 levels defined in impact height, written explicitly in the appendix of (Lewis 2009).

Line 180: it would be useful to clarify the mathematical relationship between the uncertainty estimate and the footprint. What assumptions are made to derive this relationship?

Answer, line 180: The relationship would depend on the vertical correlation function. An example: Suppose we have a profile with Gaussian uncorrelated random noise  $\varepsilon$ , such that  $\langle \varepsilon \rangle = 0$  and  $\langle \varepsilon(z)\varepsilon(z') \rangle = \sigma^2 \delta(z - z')$ . If we smooth this profile with a Gaussian filter of half width L, we get for a given altitude level,  $z_i$ , a vertically correlated error

 $\varepsilon_i = \frac{1}{L\sqrt{2\pi}} \int_{-\infty}^{\infty} \exp(-(z-z_i)/2L^2)\varepsilon(z)dz$ . If we now calculate the variance of  $\varepsilon_i$  we get  $\sigma_i^2 = \frac{\sigma^2}{L\sqrt{4\pi}}$ , which scales as 1/L.

155 Line 199: I believe what is being stated here is that G3CH is incorrectly assigning collocation error to RS92 error. We expect then, that if ERA5 is collocated to RS92 rather than RO, the RO would show the large uncertainty. Is that the case? Answer, line 199: Yes that is true, for the raw uncertainty estimates.

Line 218: is there a way to justify this interpretation using a mathematical model and showing it mathematically? Otherwise, it's difficult to assess the validity of this interpretation.

- 160 Answer, line 218: We can define the procedure of letting the area go to zero and define the collocation corrected error covariance matrix as the error covariance matrix in that limit in that limit. We have added this to section 1.2: "We are able to remove  $\varepsilon^{C}$  and the  $\varepsilon^{X}$  components of the three data sets, by adding the following additional analysis steps to the G3CH. The  $\varepsilon^{C}$  covariance matrix,  $\mathbf{C}^{C}$ , is eliminated by first calculating G3CH estimated covariance matrices  $\mathbf{C}_{i}$  for a series of collocation subsets, sampled from areas of decreasing size around th RO reference coordinates. Next, the sequence of decreasing covari-
- 165 ance estimates is extrapolated to the virtual zero-area case  $\mathbf{C}_0$ .  $\mathbf{C}_i^C = \mathbf{C}_i \mathbf{C}_0$ . Subsequently the  $\varepsilon^X$  covariance matrix,  $\mathbf{C}^X$ , is eliminated by smooting all three data sets such that they have the same vertical footprint, and then calculate for each data set a covariance matrix  $\mathbf{C}_s$  with G3CH from the smoothed data sets.  $\mathbf{C}^X = \mathbf{C}_0 \mathbf{C}_s$ . So the observation error covariance matrices that we estimate in the end includes only measurement error  $\varepsilon^I$  and representativeness error  $\varepsilon^R$ ."

Line 239: please refer back to the equations where this error covariance is defined. See the earlier comment about how the errors
break down into the different components. One could, for example, insert those components into the covariance equations (7) and identify specific outcomes depending on the properties of these error components.

Answer, line 239: We can unfortunately not refer directly to eq.7, since the presented matrices have been collocation-corrected after the G3CH was applied.

Line 270: see question raised earlier of how "footprint of the truth" is defined.

175 **Answer, line 270:** We have removed this sentence: "In the derivation of G3CH representativeness is defined with reference to given scales in space and time of the truth."

Line 275: the concept of "physical variability" is introduced here for the first time. How does it relate to the error components I, R, C, X defined earlier? Or is it a new component of error? In general, the statistical properties of the error distributions are not explicitly described (are they gaussian?) except that they are mean zero. If statistical error distribution is not relevant, and any mean-zero error distribution is acceptable, it should be stated.

180 any n

Answer, line 275: Physical variability refers to the underlying truth, which has "physical" correlations in it; temperature at altitude  $z_i$  is correlated with temperature at altitude  $z_j$ . This is not to be confused with error correlations.

Answer, line 275: The theory applies for any zero mean distribution. There is probably some obscure counterexample which from a mathematical point of view breaks this clause, so please read it as "any reasonable zero mean distribution". We have added this sentence on line 127 "Besides this no assumptions are made about the particular shape of error distribution func-

185 added tions."

Line 295: The Rieckh paper uses RO, radiosondes and analyses and forecasts. Please be more explicit why these data sets are not suitable for 3CH analysis, since they appear to be similar to the data sets used in this paper.

Answer, line 295: None of the many refractivity uncertainty experiments they do are really free from error cross correlations
between the data sets because they either use two models (which have assimilated the same observations) or radiosondes and model analysis in the same triplets. We have changed the formulation (a bit simplified, since we cannot go into a deep discussion of the Rieckh paper): "... are less well suited for 3CH analysis than the data sets used here, because the errors of the used ERA5 analysis fields must be expected to be correlated with errors of other data sets"

**References**

195 Semane, N., Anthes, R., Sjoberg, J., Healy, S., and Ruston, B.: Comparison of Desroziers and Three-Cornered Hat Methods for Estimating COSMIC-2 Bending Angle Uncertainties, Journal of Atmospheric and Oceanic Technology, 39, 929–939, https://doi.org/10.1175/JTECH-D-21-0175.1, 2022.

---

## Referee Report (RR1)

Review Summary:

The authors have diligently addressed the comments of this reviewer. The paper is ready for publication. The following is suggested:

Line 73: "with respect to a given but unknown finite vertical footprint…"

Line 108: "in the end include only…"

Line 201: insert the reference (Lewis, 2009) after "common grid".